# The Discrete Gaussian for Differential Privacy

**Clément L. Canonne**
IBM Research, Almaden
ccanonne@cs.columbia.edu

**Gautam Kamath**
University of Waterloo
g@csail.mit.edu

**Thomas Steinke**
IBM Research, Almaden
dgauss@thomas-steinke.net

## Abstract

A key tool for building differentially private systems is adding Gaussian noise to the output of a function evaluated on a sensitive dataset. Unfortunately, using a continuous distribution presents several practical challenges. First and foremost, finite computers cannot exactly represent samples from continuous distributions, and previous work has demonstrated that seemingly innocuous numerical errors can entirely destroy privacy. Moreover, when the underlying data is itself discrete (e.g., population counts), adding continuous noise makes the result less interpretable.

With these shortcomings in mind, we introduce and analyze the discrete Gaussian in the context of differential privacy. Specifically, we theoretically and experimentally show that adding discrete Gaussian noise provides essentially the same privacy and accuracy guarantees as the addition of continuous Gaussian noise. We also present an simple and efficient algorithm for exact sampling from this distribution. This demonstrates its applicability for privately answering counting queries, or more generally, low-sensitivity integer-valued queries.

## 1 Introduction

Differential Privacy [DMNS06] provides a rigorous standard for ensuring that the output of an algorithm does not leak the private details of individuals contained in its input. A standard technique for ensuring differential privacy is to evaluate a function on the input and then add a small amount of random noise to the result before releasing it. Specifically, it is common to add noise drawn from a Laplace or Gaussian distribution, which is scaled according to the *sensitivity* of the function – i.e., how much one person's data can change the function value. These are two of the most fundamental algorithms in differential privacy, which are used as subroutines in almost all differentially private systems. For example, differentially private algorithms for convex empirical risk minimization and deep learning are based on adding noise to gradients [BST14; ACGMMTZ16].

However, the Laplace and Gaussian distributions are both continuous over the real numbers. As such, it is not possible to even represent a sample from them on a finite computer, much less produce such a sample. One might suppose that such issues are purely of theoretical interest, and that they can be resolved in practice by simply using standard floating-point arithmetic and representations. Unfortunately, this is not the case: Mironov [Mir12] demonstrated that the naïve use of finite-precision approximations can result in catastrophic failures of privacy. In particular, by examining the low-order bits of the noisy output, the noiseless value can often be determined. Mironov showed that this information can allow the entire input dataset to be rapidly reconstructed, while only a negligible privacy loss is recorded by the system. Despite this demonstration, the flawed methods continue to appear in open source implementations of differentially private mechanisms [Cen18; Dif19; Goo20; Whi20]. This demonstrates a real need for us to provide safe and practical solutions to enable the deployment of differentially private systems in real-world privacy-critical settings. In this work, we

carefully consider how to securely implement these basic differentially private methods on finite computers that cannot faithfully represent real numbers.

One solution is to instead sample from a *discrete* distribution that can be sampled on a finite computer. For many natural queries, the output of the function to be computed is naturally discrete – e.g., counting how many records in a dataset satisfy some predicate – and hence there is no loss in accuracy when adding discrete noise to it. Otherwise, the function value must be rounded before adding noise.

The discrete Laplace distribution (a.k.a. two-sided geometric distribution) [GRS12] is the natural discrete analogue of the continuous Laplace distribution. That is, instead of a probability density of $\frac{\varepsilon}{2} \cdot e^{-\varepsilon|x|}$ at $x \in \mathbb{R}$ we have a probability mass of $\frac{e^\varepsilon - 1}{e^\varepsilon + 1} \cdot e^{-\varepsilon|x|}$ at $x \in \mathbb{Z}$. Like its continuous counterpart, the discrete Laplace distribution provides pure $(\varepsilon, 0)$-differential privacy and has many other desirable properties. Notably, the discrete Laplace distribution is used in the `TopDown` algorithm being developed to protect the data collected in the 2020 US Census [KCKHM18; Abo18; Cen18].

The (continuous) Gaussian distribution has many advantages over the (continuous) Laplace distribution (and also some disadvantages), making it better suited for many applications. For example, the Gaussian distribution has lighter tails than the Laplace distribution. In settings with a high degree of composition – i.e., answering many queries with independent noise, rather than a single query – the scale (e.g., variance) of Gaussian noise is also lower than the scale of Laplace noise required for a comparable privacy guarantee. The privacy analysis under composition of Gaussian noise addition is typically simpler and sharper; in particular, these privacy guarantees can be cleanly expressed in terms of *concentrated differential privacy* (CDP) [DR16; BS16] and related variants of differential privacy [Mir17; BDRS18; DRS19]. (See Section 3.1 for further discussion.)

Thus, it is natural to wonder whether a discretization of the Gaussian distribution retains the privacy and utility properties of the continuous Gaussian distribution, as is the case for the Laplace distribution. In this paper, we show that this is indeed the case.

**Definition 1** (Discrete Gaussian). *Let $\mu, \sigma \in \mathbb{R}$ with $\sigma > 0$. The discrete Gaussian distribution with location $\mu$ and scale $\sigma$ is denoted $\mathcal{N}_{\mathbb{Z}}(\mu, \sigma^2)$. It is a probability distribution supported on the integers and defined by $\forall x \in \mathbb{Z}$, $\mathbb{P}_{X \leftarrow \mathcal{N}_{\mathbb{Z}}(\mu, \sigma^2)}[X = x] = \frac{e^{-(x-\mu)^2/2\sigma^2}}{\sum_{y \in \mathbb{Z}} e^{-(y-\mu)^2/2\sigma^2}}$.*

Note that we exclusively consider $\mu \in \mathbb{Z}$; in this case, the distribution is symmetric and centered at $\mu$. This is the natural discrete analogue of the continuous Gaussian (which has density $(1/\sqrt{2\pi\sigma^2}) \cdot e^{-(x-\mu)^2/2\sigma^2}$ at $x \in \mathbb{R}$), and it arises in lattice-based cryptography (in a multivariate form, which is believed to be hard to sample from) [GPV08; Reg09; Pei10; Ste17, etc.].

## 1.1 Overview of Our Results

Our investigations focus on three aspects of the discrete Gaussian: *privacy*, *utility*, and *sampling*. In summary, we demonstrate that the discrete Gaussian provides the same level of privacy and utility as the continuous Gaussian. We also show that it can be efficiently sampled on a finite computer, thus addressing the shortcomings of continuous distributions discussed earlier.

**§2 Privacy.** The discrete Gaussian enjoys privacy guarantees which are almost identical to those of the continuous Gaussian. More precisely, in Theorem 4, we show that adding noise drawn from $\mathcal{N}_{\mathbb{Z}}(0, 1/\varepsilon^2)$ to an integer-valued sensitivity-1 query (e.g., a counting query) provides $\frac{1}{2}\varepsilon^2$-concentrated differential privacy. This is exactly the same guarantee attained by adding a draw from $\mathcal{N}(0, 1/\varepsilon^2)$. Furthermore, in Theorem 6, we provide tight bounds on the discrete Gaussian's approximate differential privacy guarantees. For large scales $\sigma$, the discrete and continuous Gaussian have virtually the same privacy guarantee. Along the way, we develop new tools for converting concentrated differential privacy guarantees into approximate differential privacy guarantees, which are not specific to the discrete Gaussian and are of independent interest.

**§3 Utility.** The accuracy attained by the discrete Gaussian is the same as (or slightly better than) the analogous continuous Gaussian. Specifically, Corollary 9 shows that the variance of $\mathcal{N}_{\mathbb{Z}}(0, \sigma^2)$ is at most $\sigma^2$, and that it also satisfies sub-Gaussian tail bounds comparable to $\mathcal{N}(0, \sigma^2)$. We show numerically that the discrete Gaussian is better than rounding the continuous Gaussian to an integral value. We also provide a thorough comparison between the discrete Gaussian and the discrete Laplace distribution in Section 3.1, with a particular focus on performance under composition.

**§4 Sampling.** We present a practical, simple, and efficient procedure for exact sampling from $\mathcal{N}_{\mathbb{Z}}\left(0, \sigma^2\right)$ that only requires access to uniformly random bits and does not involve any real-arithmetic operations or non-trivial function evaluations (Algorithm 1). As there are prior efficient methods [Kar16], we do not consider this to be one of our primary contributions. Nonetheless, we include these results for completeness and because they are arguably simpler than prior work.

On a technical note, while the takeaway of many of our conclusions is that the discrete and continuous Gaussian are qualitatively similar, we comment that such statements are non-trivial to prove, relying upon methods such as the Poisson summation formula and Fourier analysis. For instance, even basic statements on the stability property of Gaussians under linear combinations do not hold for the discrete counterpart, with approximate versions being highly involved to establish (see, e.g., [AR16]).

## 1.2 Related Work

As originally observed and demonstrated by Mironov [Mir12], naïve implementations of the Laplace mechanism with floating-point arithmetic blatantly fail to ensure differential privacy, or any form of privacy at all. As a remedy, Mironov introduced the snapping mechanism. The snapping mechanism performs rounding and truncation on top of the floating-point arithmetic implementation of Laplace noise. However, properly implementing and analyzing the snapping mechanism can be involved [Cov19], due to the idiosyncrasies of floating-point arithmetic. Furthermore, the snapping mechanism requires a compromise on privacy and accuracy, relative to what is theoretically achievable. Our methods avoid floating-point arithmetic entirely and do not compromise the privacy or accuracy guarantees. Gazeau, Miller, and Palamidessi [GMP16] gave an alternate and more general analysis of Mironov's approach of rounding the output of an inexact sampling procedure.

Ghosh, Roughgarden, and Sundararajan [GRS12] proposed and analyzed a discrete version of the Laplace mechanism, which is also private on finite computers. However, this has heavier tails than the Gaussian and requires the addition of more noise (i.e., higher variance) than the Gaussian in settings with a high degree of composition (i.e., many queries). We provide a detailed discussion in Section 3.1. The US Census Bureau intends to use discrete Laplace noise for the 2020 Census [KCKHM18; Abo18]. However, their prototype [Cen18] does not use an exact sampling procedure.

Perhaps the closest distribution to the discrete Gaussian that has been considered for privacy is the Binomial distribution. Dwork, Kenthapadi, McSherry, Mironov, and Naor [DKMMN06] gave a differential privacy analysis of Binomial noise addition, which was improved by Agarwal, Suresh, Yu, Kumar, and McMahan [ASYKM18].[1] The advantage of the Binomial is that it is amenable to distributed generation – i.e., a sum of Binomials with the same bias parameter is also Binomial. The disadvantage of Binomial noise addition, however, is that its privacy analysis is quite involved. One inherent reason for this is that the analysis must compare the Binomial to a shifted Binomial, and these distributions have different supports. If the observed output $y$ is in the support of $M(x)$ but not of $M(x')$ (i.e., $\mathbb{P}\left[M(x') = y\right] = 0$), then the privacy loss is infinite; this failure probability must be accounted for by the $\delta$ parameter of approximate $(\varepsilon, \delta)$-differential privacy. In other words, the Binomial mechanism is inherently an *approximate* differential privacy one (versus the stronger concentrated differential privacy of the discrete Gaussian). For large values of $n$, $\mathsf{Binomial}(n, p)$ provides guarantees comparable to $\mathcal{N}(0, np(1-p))$ or $\mathcal{N}_{\mathbb{Z}}\left(0, np(1-p)\right)$. This matches the intuition, since $\mathsf{Binomial}(n, p)$ converges to a Gaussian as $n \to \infty$ by the central limit theorem.

A concurrent and independent work [Goo20] analyzed what is, effectively, a truncated version of the discrete Gaussian. That work provides an almost identical sampling procedure, but a very different privacy analysis. In particular, it shows that the truncated discrete Gaussian is close to a Binomial distribution, which is, in turn, close to a rounded Gaussian. And the privacy analysis is based on this closeness. Our analysis is more direct.

Going beyond noise addition, it has been shown that private histograms [BV17] and selection (i.e., the exponential mechanism) [Ilv19] can be implemented on finite computers. (Both of these results are for pure $(\varepsilon, 0)$-differential privacy.) We remark that our noise addition methods can also form the basis of an implementation of these methods. For example, instead of the exponential mechanism, we can implement the "Report Noisy Max" algorithm [DR14], which uses Laplace or Exponential or Gumbel [Ada13] noise to perform the same task of selection.

To the best of our knowledge, there has been no work on implementing an analogue of Gaussian noise addition on finite computers. An obvious approach would be to round the output of some inexact sampling procedure. Properly analyzing this may be difficult since the underlying inexact Gaussian sampling procedure will be more complex than the equivalent for Laplace. Furthermore, in Section 3, we show empirically that our approach yields better utility than rounding.

In Proposition 7, we give a conversion from Rényi and concentrated differential privacy to approximate differential privacy. Asoodeh, Liao, Calmon, Kosut, and Sankar [ALCKS20] provide an optimal conversion from Rényi differential privacy to approximate differential privacy as well as some approximations that subsume ours. Their optimal result is, by definition, tighter than ours (but only slightly) at the expense of being more complicated and less numerically stable. See Section 2.3.

Another of our secondary contributions is a simple and efficient method for sampling from a discrete Gaussian or discrete Laplace; see Section 4. Karney [Kar16] and Du, Fan, and Wei [DFW20] also provide such algorithms. We consider our method to be simpler. In particular, our method keeps all arithmetic within the integers or the rational numbers, where exact arithmetic is possible. In contrast, Karney's method still involves representing real numbers, but this can be carefully implemented on a finite computer using a flexible level of precision and lazy evaluation – that is, although a uniform sample from $[0, 1]$ requires an infinite number of bits to represent, only a finite (but a priori unbounded) number of these bits are actually needed and these can be sampled when needed. There are also methods for *approximate* sampling [ZSS19], but our interest is in exact sampling.

Finally, we remark that (a multivariate version of) the discrete Gaussian has been extensively studied in the context of lattice-based cryptography [GPV08; Reg09; Pei10; Ste17, etc.].

## 2 Privacy

For completeness, we state the definitions of differential privacy [DMNS06; DKMMN06] and concentrated differential privacy [DR16; BS16]. We will then show that the discrete and continuous Gaussians provide nearly identical guarantees under both definitions.

**Definition 2** (Pure/Approximate Differential Privacy). *A randomized algorithm $M \colon \mathcal{X}^n \to \mathcal{Y}$ satisfies $(\varepsilon, \delta)$-differential privacy if, for all $x, x' \in \mathcal{X}^n$ differing on a single element and all events $E \subset \mathcal{Y}$, we have $\mathbb{P}\left[M(x) \in E\right] \le e^{\varepsilon} \cdot \mathbb{P}\left[M(x') \in E\right] + \delta$.*

The special case of $(\varepsilon, 0)$-differential privacy is referred to as *pure* or *pointwise $\varepsilon$*-differential privacy, whereas, for $\delta > 0$, $(\varepsilon, \delta)$-differential privacy is referred to as *approximate differential privacy*.

**Definition 3** (Concentrated Differential Privacy). *A randomized algorithm $M \colon \mathcal{X}^n \to \mathcal{Y}$ satisfies $\frac{1}{2}\varepsilon^2$-concentrated differential privacy if, for all $x, x' \in \mathcal{X}^n$ differing on a single element and all $\alpha \in (1, \infty)$, we have $\mathrm{D}_{\alpha}\left(M(x) \| M(x')\right) \le \frac{1}{2}\varepsilon^2\alpha$, where $\mathrm{D}_{\alpha}\left(P \| Q\right) = \frac{1}{\alpha-1}\log\sum_{y \in \mathcal{Y}} P(y)^{\alpha}Q(y)^{1-\alpha}$ is the Rényi divergence of order $\alpha$ of the distribution $P$ from the distribution $Q$.*[2]

Note that $(\varepsilon, 0)$-differential privacy implies $\frac{1}{2}\varepsilon^2$-concentrated differential privacy, which, in turn, implies $\left(\frac{1}{2}\varepsilon^2 + \varepsilon \cdot \sqrt{2\log(1/\delta)}, \delta\right)$-differential privacy for all $\delta > 0$ [BS16].

### 2.1 Concentrated Differential Privacy

**Theorem 4** (Discrete Gaussian Satisfies Concentrated Differential Privacy). *Let $\Delta, \varepsilon > 0$. Let $q \colon \mathcal{X}^n \to \mathbb{Z}$ satisfy $|q(x) - q(x')| \le \Delta$ for all $x, x' \in \mathcal{X}^n$ differing on a single entry. Define a randomized algorithm $M \colon \mathcal{X}^n \to \mathbb{Z}$ by $M(x) = q(x) + Y$ where $Y \leftarrow \mathcal{N}_{\mathbb{Z}}\left(0, \Delta^2/\varepsilon^2\right)$. Then $M$ satisfies $\frac{1}{2}\varepsilon^2$-concentrated differential privacy.*

The continuous Gaussian satisfies exactly the same concentrated differential privacy bound. To prove Theorem 4, we use the following well-known (e.g., [Reg09]) technical lemma.

**Lemma 5.** *Let $\mu, \sigma \in \mathbb{R}$ with $\sigma > 0$. Then $\sum_{x \in \mathbb{Z}} e^{-(x-\mu)^2/2\sigma^2} \le \sum_{x \in \mathbb{Z}} e^{-x^2/2\sigma^2}$.*

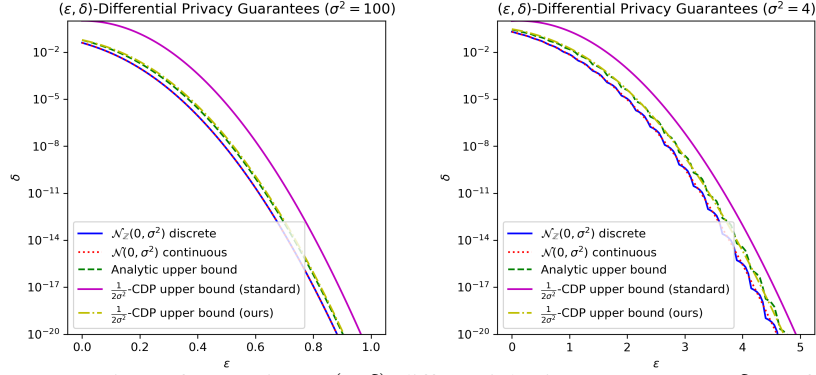

Figure 1: Comparison of approximate $(\varepsilon, \delta)$-differential privacy guarantees ($\delta$ as a function of $\varepsilon$).

*Proof.* Let $f\colon \mathbb{R} \to \mathbb{R}$ be defined by $f(x) = e^{-x^2/2\sigma^2}$. Define its Fourier transform $\hat{f}\colon \mathbb{R} \to \mathbb{R}$ by $\hat{f}(y) = \int_{\mathbb{R}} f(x)e^{-2\pi\sqrt{-1}xy}\mathrm{d}x = \sqrt{2\pi\sigma^2} \cdot e^{-2\pi^2\sigma^2 y^2}$. By the Poisson summation formula [Poi; Wei], for every $t \in \mathbb{R}$, we have $\sum_{x\in\mathbb{Z}} f(x+t) = \sum_{y\in\mathbb{Z}} \hat{f}(y) \cdot e^{2\pi\sqrt{-1}yt}$. (This is the Fourier series representation of the $1$-periodic function $g\colon \mathbb{R} \to \mathbb{R}$ given by $g(t) = \sum_{x\in\mathbb{Z}} e^{-(x+t)^2/2\sigma^2}$.) In particular, $f(x) > 0$ and $\hat{f}(x) > 0$ for all $x \in \mathbb{R}$. From this and the triangle inequality, we get $\sum_{x\in\mathbb{Z}} e^{-(x-\mu)/2\sigma^2} = \sum_{x\in\mathbb{Z}} f(x-\mu) = |\sum_{x\in\mathbb{Z}} f(x-\mu)| = |\sum_{y\in\mathbb{Z}} \hat{f}(y)e^{-2\pi\sqrt{-1}y\mu}| \leq \sum_{y\in\mathbb{Z}} |\hat{f}(y)e^{-2\pi\sqrt{-1}y\mu}| = \sum_{y\in\mathbb{Z}} \hat{f}(y) = \sum_{x\in\mathbb{Z}} f(x) = \sum_{x\in\mathbb{Z}} e^{-x^2/2\sigma^2}$, proving the lemma. $\square$

*Proof of Theorem 4.* For all $\mu \in \mathbb{Z}$ and $\alpha \in (1, \infty)$ and $\sigma^2 > 0$, we have

$$e^{(\alpha-1)\mathrm{D}_\alpha\left(\mathcal{N}_{\mathbb{Z}}\left(\mu,\sigma^2\right)\big\|\mathcal{N}_{\mathbb{Z}}\left(0,\sigma^2\right)\right)} = \sum_{x\in\mathbb{Z}} \mathop{\mathbb{P}}_{X\leftarrow\mathcal{N}_{\mathbb{Z}}(\mu,\sigma^2)}\big[X = x\big]^\alpha \cdot \mathop{\mathbb{P}}_{X\leftarrow\mathcal{N}_{\mathbb{Z}}(0,\sigma^2)}\big[X = x\big]^{1-\alpha}$$

$$= \sum_{x\in\mathbb{Z}} \left(\frac{e^{-(x-\mu)^2/2\sigma^2}}{\sum_{y\in\mathbb{Z}} e^{-(y-\mu)^2/2\sigma^2}}\right)^\alpha \cdot \left(\frac{e^{-x^2/2\sigma^2}}{\sum_{y\in\mathbb{Z}} e^{-y^2/2\sigma^2}}\right)^{1-\alpha} = \frac{\sum_{x\in\mathbb{Z}} \exp\left(\frac{-x^2+2\alpha\mu x-\alpha\mu^2}{2\sigma^2}\right)}{\sum_{y\in\mathbb{Z}} e^{-y^2/2\sigma^2}}$$

$$= e^{\alpha(\alpha-1)\mu^2/2\sigma^2} \cdot \frac{\sum_{x\in\mathbb{Z}} e^{-(x-\alpha\mu)^2/2\sigma^2}}{\sum_{y\in\mathbb{Z}} e^{-y^2/2\sigma^2}} \leq e^{\alpha(\alpha-1)\mu^2/2\sigma^2},$$

where the final inequality follows from Lemma 5; the inequality is an equality when $\alpha\mu \in \mathbb{Z}$. Thus $\mathrm{D}_\alpha\left(\mathcal{N}_{\mathbb{Z}}\left(\mu,\sigma^2\right)\big\|\mathcal{N}_{\mathbb{Z}}\left(\nu,\sigma^2\right)\right) = \mathrm{D}_\alpha\left(\mathcal{N}_{\mathbb{Z}}\left(\mu-\nu,\sigma^2\right)\big\|\mathcal{N}_{\mathbb{Z}}\left(0,\sigma^2\right)\right) \leq \frac{(\mu-\nu)^2}{2\sigma^2} \cdot \alpha$, which implies the result by Definition 3, since $M(x) \sim \mathcal{N}_{\mathbb{Z}}\left(q(x),\sigma^2\right)$. $\square$

## 2.2 Approximate Differential Privacy

In the full version, we prove the following tight approximate differential privacy bound.

**Theorem 6** (Discrete Gaussian Satisfies Approximate Differential Privacy). *Let $\Delta, \sigma, \varepsilon > 0$. Let $q\colon \mathcal{X}^n \to \mathbb{Z}$ satisfy $|q(x) - q(x')| \leq \Delta$ for all $x, x' \in \mathcal{X}^n$ differing on a single entry. Define a randomized algorithm $M\colon \mathcal{X}^n \to \mathbb{Z}$ by $M(x) = q(x)+Y$ where $Y \leftarrow \mathcal{N}_{\mathbb{Z}}\left(0,\sigma^2\right)$. Then $M$ satisfies $(\varepsilon, \delta)$-differential privacy for $\delta = \mathop{\mathbb{P}}_{Y\leftarrow\mathcal{N}_{\mathbb{Z}}(0,\sigma^2)}\left[Y > \frac{\varepsilon\sigma^2}{\Delta} - \frac{\Delta}{2}\right] - e^\varepsilon \cdot \mathop{\mathbb{P}}_{Y\leftarrow\mathcal{N}_{\mathbb{Z}}(0,\sigma^2)}\left[Y > \frac{\varepsilon\sigma^2}{\Delta} + \frac{\Delta}{2}\right]$. Furthermore, this is the smallest possible value of $\delta$ for which this is true.*

If we replace all occurrences of the discrete Gaussian with the continuous Gaussian above, then the same result holds [BW18, Thm. 8]. Empirically, these guarantees are very close. In Figure 1, we empirically compare the optimal $\delta$ (given by Theorem 6) to the bound attained by the corresponding continuous Gaussian, as well as upper bounds entailed by concentrated differential privacy – the standard upper bound [BS16, Prop. 1.3] and our improved bound (Proposition 7). We see that our upper bound is reasonably tight. The discrete and continuous Gaussian attain almost identical guarantees for large $\sigma$, but discretization effects become apparent for small $\sigma$.

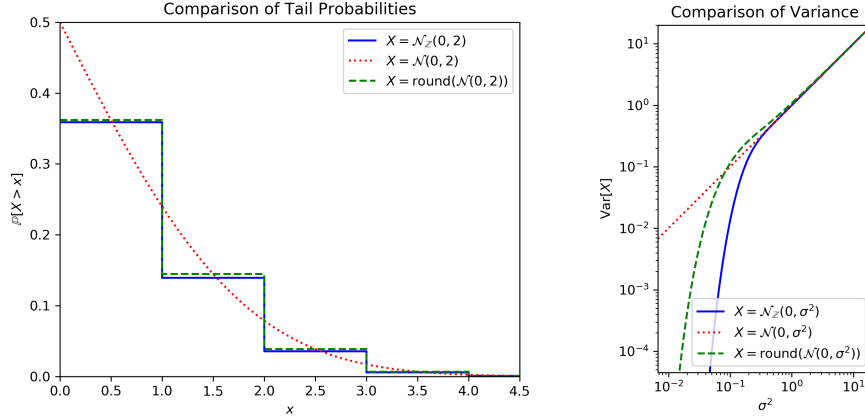

Figure 2: Comparison of tail bounds and variance for continuous, discrete, and rounded Gaussians.

## 2.3 Converting Concentrated Differential Privacy to Approximate Differential Privacy

We have stated guarantees for both concentrated differential privacy and approximate differential privacy. Now we show how to convert from the former to the latter. This is useful if the discrete Gaussian is being used repeatedly and we want to provide a privacy guarantee for the composition – concentrated differential privacy has cleaner composition guarantees than approximate differential privacy. We include this result for completeness; similar results are known [ALCKS20]. The proof can be found in the full version.

**Proposition 7.** *Let $M : \mathcal{X}^n \to \mathcal{Y}$ be a randomized algorithm satisfying $\rho$-concentrated differential privacy. Then $M$ is $(\varepsilon, \delta)$-differentially private for any $\varepsilon \geq 0$ and*

$$\delta = \inf_{\alpha \in (1,\infty)} (\alpha - 1)^{-1} e^{(\alpha-1)(\alpha\rho-\varepsilon)} \cdot (1 - 1/\alpha)^\alpha \leq \inf_{\alpha \in (1,\infty)} (\alpha - 1)^{-1} e^{(\alpha-1)(\alpha\rho-\varepsilon)-1}. \quad (1)$$

Proposition 7 should be contrasted with the standard bound [DRV10; DR16; BS16; Mir17] of $\delta = \inf_{\alpha \in (1,\infty)} e^{(\alpha-1)(\alpha\rho-\varepsilon)} = e^{-(\varepsilon-\rho)^2/4\rho}$, which holds when $\varepsilon \geq \rho > 0$. Bun and Steinke [BS16] prove an intermediate bound of $\delta = 2\sqrt{\pi\rho} \cdot e^\varepsilon \cdot \mathbb{P}_{X \leftarrow \mathcal{N}(0,1)} \left[ X > \frac{\varepsilon+\rho}{\sqrt{2\rho}} \right]$. Asoodeh, Liao, Calmon, Kosut, and Sankar [ALCKS20] provide an optimal conversion from Rényi differential privacy [Mir17] to approximate differential privacy. Taking the infimum over the divergence parameter $\alpha$, their bound yields $\delta = \inf_{\alpha \in (1,\infty)} \inf \left\{ \hat{\delta} \in [0,1] : \forall p \in (\hat{\delta}, 1) \; p^\alpha (p - \hat{\delta})^{1-\alpha} + (1-p)^\alpha (e^\varepsilon - p + \hat{\delta})^{1-\alpha} \leq e^{(\alpha-1)(\alpha\rho-\varepsilon)} \right\}$. Clearly, the expression in Proposition 7 is simpler than this. Moreover, this expression is numerically unstable for small values of $\delta$. We implemented both methods and found that they yield very similar results for the parameter regime of interest, but numerical stability was a significant practical issue.

**Efficient computation of $\delta$.** We can efficiently compute the bound of Proposition 7: the inequality in the guarantee has an analytic minimizer, while the equality is equivalent to

$$\delta = \inf_{\alpha \in (1,\infty)} e^{g(\alpha)} \quad \text{where} \quad g(\alpha) = (\alpha - 1)(\alpha\rho - \varepsilon) + \alpha \cdot \log(1 - 1/\alpha) - \log(\alpha - 1). \quad (2)$$

Since $g$ is a smooth convex function, it has a unique minimizer, which can be found by binary search.

## 3 Utility

We now consider how much noise the discrete Gaussian adds. As a comparison point, we consider both the continuous Gaussian and, in the interest of a fair comparison, the rounded Gaussian – i.e., a sample from the continuous Gaussian rounded to the nearest integral value. In Figure 2, we show how these compare numerically. We see that the tail of the rounded Gaussian stochastically dominates that of the discrete Gaussian. In other words, the utility of the discrete Gaussian is strictly better than the rounded Gaussian (although not by much for reasonable values of $\sigma$).

To obtain analytic bounds, we begin by bounding the moment generating function:

**Lemma 8.** *Let $t, \sigma \in \mathbb{R}$ with $\sigma > 0$. Then* $\underset{X \leftarrow \mathcal{N}_{\mathbb{Z}}(0,\sigma^2)}{\mathbb{E}} \left[ e^{tX} \right] \leq e^{t^2 \sigma^2 / 2}$.

The proof readily follows from Lemma 5. For comparison, recall that the continuous Gaussian satisfies the same bound, but with equality: $\underset{X \leftarrow \mathcal{N}(0,\sigma^2)}{\mathbb{E}} \left[ e^{tX} \right] = e^{t^2 \sigma^2 / 2}$ for all $t, \sigma \in \mathbb{R}$ with $\sigma > 0$.

The bound on the moment generating function shows that the discrete Gaussian is subgaussian [Riv12]. Standard facts about subgaussian random variables yield bounds on the variance and tails:

**Corollary 9.** *Let $X \leftarrow \mathcal{N}_{\mathbb{Z}} \left(0, \sigma^2\right)$. Then $\mathsf{Var}\left[X\right] \leq \sigma^2$ and $\mathbb{P}\left[X \geq \lambda\right] \leq e^{-\lambda^2/2\sigma^2}$ for all $\lambda \geq 0$.*

Thus the variance of the discrete Gaussian is at most that of the corresponding continuous Gaussian and we also have subgaussian tail bounds. In fact, it is possible to obtain slightly tighter bounds, showing that the variance of the discrete Gaussian is *strictly less* than that of the continuous Gaussian. We elaborate on this in the full version. However, these improvements are most pronounced for small $\sigma$, which is not the typical regime of interest for differential privacy.

## 3.1 Discrete Laplace

We now compare the discrete Gaussian with the most obvious alternative – the discrete Laplace. We first give a formal definition and some relevant facts. The discrete Laplace (also known as the *two-sided geometric*) was introduced into the differential privacy literature by Ghosh, Roughgarden, and Sundararajan [GRS12], who showed that it satisfies strong optimality properties.

**Definition 10** (Discrete Laplace). *Let $t > 0$. The discrete Laplace distribution with scale parameter $t$ is denoted $\mathrm{Lap}_{\mathbb{Z}}(t)$. It is a probability distribution supported on the integers and defined by* $\underset{X \leftarrow \mathrm{Lap}_{\mathbb{Z}}(t)}{\mathbb{P}} \left[X = x\right] = \frac{e^{1/t}-1}{e^{1/t}+1} \cdot e^{-|x|/t}$ *for all $x \in \mathbb{Z}$.*

**Lemma 11** (Discrete Laplace Privacy). *Let $\Delta, \varepsilon > 0$. Let $q \colon \mathcal{X}^n \to \mathbb{Z}$ satisfy $|q(x) - q(x')| \leq \Delta$ for all $x, x' \in \mathcal{X}^n$ differing on a single entry. Define a randomized algorithm $M \colon \mathcal{X}^n \to \mathbb{Z}$ by $M(x) = q(x) + Y$ where $Y \leftarrow \mathrm{Lap}_{\mathbb{Z}}(\Delta/\varepsilon)$. Then $M$ satisfies $(\varepsilon, 0)$-differential privacy.*

**Lemma 12** (Discrete Laplace Utility). *Let $\varepsilon > 0$ and let $Y \leftarrow \mathrm{Lap}_{\mathbb{Z}}(1/\varepsilon)$. The distribution is symmetric; in particular, $\mathbb{E}\left[Y\right] = 0$. We have $\mathbb{E}\left[|Y|\right] = \frac{2 \cdot e^\varepsilon}{e^{2\varepsilon}-1}$ and $\mathsf{Var}\left[Y\right] = \mathbb{E}\left[Y^2\right] = \frac{2 \cdot e^\varepsilon}{(e^\varepsilon-1)^2}$. For all $\lambda < \varepsilon$, $\mathbb{E}\left[e^{\lambda|Y|}\right] = \frac{e^\varepsilon-1}{e^\varepsilon+1} \cdot \frac{e^{\varepsilon-\lambda}+1}{e^{\varepsilon-\lambda}-1}$. For all $m \in \mathbb{N}$, $\mathbb{P}\left[Y \geq m\right] = \mathbb{P}\left[Y \leq -m\right] = \frac{e^{-\varepsilon(m-1)}}{e^\varepsilon+1}$.*

We remark that the discrete Laplace can also be efficiently sampled; see the full version.

There are two immediate qualitative differences between the discrete Laplace and the discrete Gaussian.[3] In terms of utility, the discrete Laplace has subexponential tails (i.e., decaying as $e^{-\varepsilon m}$), whereas the discrete Gaussian has subgaussian tails (i.e., decaying as $e^{-\varepsilon^2 m^2/2}$). In terms of privacy, the discrete Gaussian satisfies concentrated differential privacy, whereas the discrete Laplace satisfies pure differential privacy; pure differential privacy is a qualitatively stronger privacy condition. Thus neither distribution universally dominates the other. They offer different privacy-utility tradeoffs.

We now consider a quantitative comparison. To quantify utility, we focus on the variance of the distribution. (An alternative would be to consider the width of a confidence interval.) For now, we will quantify privacy by concentrated differential privacy. Pure $(\varepsilon, 0)$-differential privacy implies $\frac{1}{2}\varepsilon^2$-concentrated differential privacy; thus both distributions can be evaluated on this scale.

Consider a small $\varepsilon > 0$ and a counting query. We can attain $\frac{1}{2}\varepsilon^2$-concentrated differential privacy by adding noise from either $\mathcal{N}_{\mathbb{Z}}\left(0, 1/\varepsilon^2\right)$ or $\mathrm{Lap}_{\mathbb{Z}}(1/\varepsilon)$. We have

$$\frac{1}{\varepsilon^2} \geq \underset{Y_{\mathsf{G}} \leftarrow \mathcal{N}_{\mathbb{Z}}(0,1/\varepsilon^2)}{\mathsf{Var}}\left[Y_{\mathsf{G}}\right] \geq \frac{1}{e^{\varepsilon^2}-1} = \frac{1-o(1)}{\varepsilon^2} \quad \text{and} \quad \underset{Y_{\mathsf{L}} \leftarrow \mathrm{Lap}_{\mathbb{Z}}(1/\varepsilon)}{\mathsf{Var}}\left[Y_{\mathsf{L}}\right] = \frac{2 \cdot e^\varepsilon}{(e^\varepsilon-1)^2} = \frac{2 \pm o(1)}{\varepsilon^2}.$$

Thus, asymptotically (i.e., for small $\varepsilon$), the discrete Gaussian has half as much variance as the discrete Laplace for the same level of privacy. In this comparison, our approach is clearly better.

However, the above quantitative comparison is potentially unfair. Quantifying differential privacy by concentrated differential privacy may favour the Gaussian. If instead we demand pure $(\varepsilon, 0)$-differential privacy or approximate $(\varepsilon, \delta)$-differential privacy, then the comparison may yield the opposite conclusion. It is fundamentally difficult to compare different versions of differential privacy.

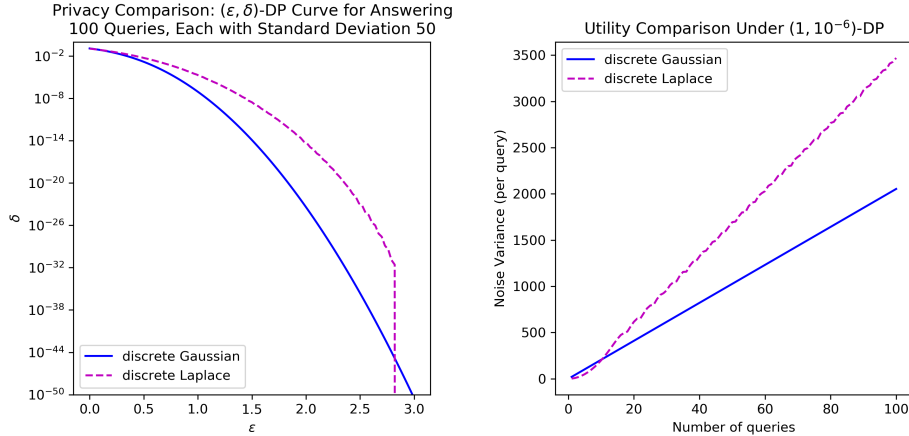

Figure 3: Comparison of discrete Gaussian and discrete Laplace noise addition. Left: Utility is fixed (i.e., answer $k = 100$ counting queries each with variance $50^2$) and we consider the curve of achievable $(\varepsilon, \delta)$-differential privacy guarantees. Right: Privacy is fixed (i.e., $(1, 10^{-6})$-differential privacy) and we consider the utility (i.e., variance) as we vary the number of queries to be answered.

There is another factor to consider: A practical differentially private system will answer many queries via independent noise addition. Thus the real object of interest is the privacy and utility of the *composition* of many applications of noise addition.

For the rest of this section, we consider the task of answering $k$ counting queries (or sensitivity-1 queries) by adding either discrete Gaussian or discrete Laplace noise. We measure privacy by approximate $(\varepsilon, \delta)$-differential privacy over a range of parameters. Results are in Figure 3.

Concentrated differential privacy has an especially clean composition theorem [BS16]: The $k$-fold composition of $\frac{1}{2}\varepsilon^2$-concentrated differential privacy satisfies $\frac{1}{2}k\varepsilon^2$-concentrated differential privacy. Thus, to attain $\frac{1}{2}\varepsilon^2$-concentrated differential privacy for $k$ counting queries, it suffices to add noise from $\mathcal{N}_{\mathbb{Z}}\left(0, k/\varepsilon^2\right)$ to each value independently. We convert this guarantee into approximate $(\varepsilon', \delta')$-differential privacy using Proposition 7. In contrast, the composition of multiple additions of discrete Laplace noise is not as clean. We use an optimal composition result provided by Kairouz, Oh, and Viswanath [KOV17; MV16]: The $k$-fold composition of $(\varepsilon, \delta)$-differential privacy satisfies $(\varepsilon', \delta')$-differential privacy if and only if $\frac{1}{(1+e^\varepsilon)^k} \sum_{\ell=0}^{k} \binom{k}{\ell} \max\left\{0, e^{\ell\varepsilon} - e^{\varepsilon'+(k-\ell)\varepsilon}\right\} \leq 1 - \frac{1-\delta'}{(1-\delta)^k}$.

Figure 3 shows that the discrete Gaussian provides a better privacy-utility tradeoff than the discrete Laplace, except in two narrow parameter regimes: Either a small number of queries ($k \leq 10$ on the right) or if we demand minuscule $\delta$ ($\delta < 10^{-45}$ on the left). We only compare variances; if we compare confidence interval sizes, then this would further advantage the lighter-tailed Gaussian.

## 4 Sampling

In Algorithm 1, we present a method to efficiently sample exactly from a discrete Gaussian on a finite computer given access only to uniformly random bits. This satisfies the guarantee in Theorem 13.

For simplicity, we focus our discussion of runtime only on the expected number of arithmetic operations; each such operation will take time polylogarithmic in the bit complexity of the parameters (e.g., in the representation of $\sigma^2$ as a rational number). High probability bounds can also be obtained: to generate $k$ draws from a discrete Gaussian, our algorithm requires $O(k + \log(1/\delta))$ arithmetic operations except with probability $\delta$. See the full version for further discussion of the runtime.

**Theorem 13.** *On input $\sigma^2 \in \mathbb{Q}$, the procedure described in Algorithm 1 outputs one sample from $\mathcal{N}_{\mathbb{Z}}\left(0, \sigma^2\right)$ and requires only a constant number of operations in expectation.*

The idea behind the algorithm is to first sample from a discrete Laplace distribution and then "convert" this into a discrete Gaussian by rejection sampling. Discrete Laplace is simply a geometric random variable with a random sign (and a rejection step to avoid double counting zero). To generate a geometric random variable, we separately sample the high order bits (by repeatedly flipping a coin

until we see a 0) and the low order bits (by rejection sampling). If $\sigma^2$ is rational, then all operations are over rational numbers. `Python` code for Algorithm 1 (and our Figures) is available online [Dga].

---

**Algorithm 1** Algorithm for Sampling a Discrete Gaussian

---

**Input:** Parameter $\sigma^2 > 0$. **Output:** One sample from $\mathcal{N}_{\mathbb{Z}}\left(0, \sigma^2\right)$.
Set $t \leftarrow \lfloor \sigma \rfloor + 1$
**loop**                                                                          ▷ Repeat until successful.
    Sample $U \in \{0, 1, 2, \cdots, t - 1\}$ uniformly at random.
    Sample $D \leftarrow \text{Bernoulli}(\exp(-U/t))$.                          ▷ Use Algorithm 2.
    If $D = 0$, then reject and restart.
    Initialize $V \leftarrow 0$.
    **loop**                                            ▷ Generate $V$ from $\text{Geometric}(1 - e^{-1})$.
        Sample $A \leftarrow \text{Bernoulli}(\exp(-1))$.
        If $A = 0$, then break the inner loop. Else set $V \leftarrow V + 1$ and continue.
    **end loop**                                         ▷ $U + t \cdot V$ is $\text{Geometric}(1 - e^{-1/t})$.
    Sample $B \leftarrow \text{Bernoulli}(1/2)$.
    If $B = 1$ and $U = 0$ and $V = 0$, then reject and restart.
    Set $Z \leftarrow (1 - 2B) \cdot (U + t \cdot V)$.                      ▷ Now $Z$ is a discrete Laplace.
    Sample $C \leftarrow \text{Bernoulli}(\exp(-(|Z| - \sigma^2/t)^2/2\sigma^2)))$.
    If $C = 0$, then reject and restart. Else **return** $Z$ as output. ▷ Success; $Z$ is a discrete Gaussian.
**end loop**

---

---

**Algorithm 2** Algorithm for Sampling $\text{Bernoulli}(\exp(-\gamma))$.

---

**Input:** Parameter $\gamma \geq 0$. **Output:** One sample from $\text{Bernoulli}(\exp(-\gamma))$.
**if** $\gamma \in [0, 1]$ **then**
    Set $K \leftarrow 1$.
    **loop**
        Sample $A \leftarrow \text{Bernoulli}(\gamma/K)$.
        If $A = 0$, then break the loop.
        If $A = 1$, then set $K \leftarrow K + 1$ and continue the loop.
    **end loop**
    If $K$ is odd, then **return** 1.
    If $K$ is even, then **return** 0.
**else**
    **for** $k = 1$ **to** $\lfloor \gamma \rfloor$ **do**
        Sample $B \leftarrow \text{Bernoulli}(\exp(-1))$                         ▷ Recursive call.
        If $B = 0$, then break the loop and **return** 0.
    **end for**
    Sample $C \leftarrow \text{Bernoulli}(\exp(\lfloor \gamma \rfloor - \gamma))$           ▷ Recursive call. $\gamma - \lfloor \gamma \rfloor \in [0, 1]$.
    **return** $C$.
**end if**

---

Algorithm 1 requires sampling from $\text{Bernoulli}(\exp(-\gamma))$ as a subroutine; we show how to do this in Algorithm 2. We reduce the task of sampling from $\text{Bernoulli}(\exp(-\gamma))$ to that of sampling from $\text{Bernoulli}(\gamma/k)$ for various integers $k \geq 1$. This procedure is based on a technique of von Neumann [VN51; For72]. This procedure avoids complex operations, such as computing the exponential function. Thus, for a rational $\gamma$, this can be implemented on a finite computer. Specifically, for $n, d \in \mathbb{N}$, to sample $\text{Bernoulli}(n/d)$ it suffices to draw $D \in \{1, 2, \ldots, d\}$ uniformly at random and output 1 if $D \leq n$ and output 0 if $D > n$. (To sample $D \in \{1, 2, \cdots, d\}$ we can again use rejection sampling – that is, we uniformly sample $D \in \{1, 2, \cdots, 2^{\lceil \log_2 d \rceil}\}$ and reject and retry if $D > d$.)

Algorithm 2 attains the following guarantee. (We take sampling $\text{Bernoulli}(n/d)$ given $n, d \in \mathbb{N}$ to require a constant number of arithmetic operations in expectation.)

**Proposition 14.** *On input (rational) $\gamma \geq 0$, the procedure described in Algorithm 2 outputs on sample from* $\text{Bernoulli}(\exp(-\gamma))$, *and requires a constant number of operations in expectation.*

## Broader Impact

We have provided a thorough analysis of the privacy and utility properties of the discrete Gaussian and the practicality of sampling it. The impact of this work is that it makes the real-world deployment of differential privacy more practical and secure. In particular, we bridge the gap between the theory (which considers continuous distributions) and the practice (where precision is finite and numerical errors can cause a dramatic privacy failures). We hope that the discrete Gaussian will be used in practice and, further, that our work is critical to enabling these real-world deployments.

The positive impact of this work is clear: Differential privacy provides a principled and quantitative way to balance rigorous privacy guarantees and statistical utility in data analysis. If this technology is adopted, it can provide untrusted third parties controlled access to data (e.g., to enable scientific research), while affording the data subjects (i.e., the general public) an adequate level of privacy protection. In any case, our methods are better than using flawed methods (i.e., naïve floating-point implementations of continuous distributions) that inject noise without actually protecting privacy or using methods (such as rounding or discrete Laplace) that offer a worse privacy-utility tradeoff.

The negative impact of this work is less clear. All technologies can be misused. For example, an organization may be able to deceptively claim that their system protects privacy on the basis that it is differentially private, when, in reality, it is not private at all, because their privacy parameter is enormous (e.g., $\varepsilon = 10^6$). One needs to be careful and critical about promises made by such companies, and educate the general audience about what differential privacy does provide, what it does not, and when guarantees end up being meaningless.

However, we must acknowledge that there is a small – but vocal – group of people who do not want differential privacy to be deployed in practice. In particular, the US Census Bureau's planned adoption of differential privacy for the 2020 US Census has met staunch opposition from some social scientists. We cannot speak for the opponents of differential privacy; many of their objections do not make sense to us and thus it would be inappropriate for us to try summarizing them. However, there is a salient point that needs to be discussed:

Differential privacy provides a principled and quantitative way to balance rigorous privacy guarantees and statistical utility in data analysis. This is good, in theory, but, in practice, privacy versus utility is a heated and muddy debate. On one hand, data users (such as social scientists) want unfettered access to the raw data. On the other hand, privacy advocates want the data locked up or never collected in the first place. The technology of differential privacy offers a vehicle for compromise. Yet, some parties are not interested in compromise. In particular, users of census data users are accustomed to largely unrestricted data access. From a privacy perspective, this is unsustainable – the development of reconstruction attacks and the availability of large auxiliary datasets for linking/re-identification has shown that census data needs more robust protections. Understandably, those who rely on census data are deeply concerned about anything that may compromise their ability to conduct research. The adoption of differential privacy has prompted uncomfortable (but necessary) discussions about the value of providing data access relative to the privacy cost. In particular, it is necessary to decide how to allocate the privacy budget – which statistics are most important to release accurately?

Another dimension of the privacy-versus-utility debate is how it affects small communities, such as racial/ethnic minorities or rural populations. Smaller populations inherently suffer a harsher privacy-utility tradeoff. Differential privacy is almost always defined so that it provides every person with an equal level of privacy. Consequently, differentially private statistics for smaller populations (e.g., Native Americans in a small settlement) will be less accurate than for larger populations (e.g., Whites in a large US city). More precisely, noise addition methods like ours offer the same absolute accuracy on all populations, but the relative accuracy will be worse when the denominator (i.e., population size) is smaller. The only alternative is to offer small communities weaker privacy protections. We stress that this issue is *not* specific to differential privacy. For example, if we rely on anonymity or de-identification, then we must grapple with the fact that minorities are more easily re-identified, since, by definition, minorities are more unique. This is a fundamental tradeoff that needs to be carefully considered with input from the minorities and communities concerned.

Ultimately, computer scientists can only provide tools and it is up to policymakers in government and other organizations to decide how to use them. This work, along with the broader literature on differential privacy, provides such tools. However, the research community also has a responsibility to provide instructions for how these tools should and should not be used.

## Acknowledgments and Disclosure of Funding

We thank Shahab Asoodeh, Damien Desfontaines, Peter Kairouz, and Ananda Theertha Suresh for making us aware of several related works.

GK was supported by an NSERC Discovery grant, a Compute Canada RRG grant, and a University of Waterloo startup grant. CC and TS were directly supported by IBM. TS is now at Google.

## Footnotes

[0]Alphabetical authorship. Full version available at https://arxiv.org/abs/2004.00010

[1] Dinur and Nissim [DN03] also analyzed the privacy properties of Binomial noise addition, but this predates the definition of differential privacy, so they work with a different definition.

[2]We take log to be the natural logarithm – i.e., base $e \approx 2.718$.

[3] The entire discussion in this section applies equally well to the continuous analogues of these distributions.

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
