[Supplementary Material]

# The Discrete Gaussian for Differential Privacy

Clément L. Canonne[*]    Gautam Kamath[†]    Thomas Steinke[‡]

## Abstract

A key tool for building differentially private systems is adding Gaussian noise to the output of a function evaluated on a sensitive dataset. Unfortunately, using a continuous distribution presents several practical challenges. First and foremost, finite computers cannot exactly represent samples from continuous distributions, and previous work has demonstrated that seemingly innocuous numerical errors can entirely destroy privacy. Moreover, when the underlying data is itself discrete (e.g., population counts), adding continuous noise makes the result less interpretable.

With these shortcomings in mind, we introduce and analyze the discrete Gaussian in the context of differential privacy. Specifically, we theoretically and experimentally show that adding discrete Gaussian noise provides essentially the same privacy and accuracy guarantees as the addition of continuous Gaussian noise. We also present an simple and efficient algorithm for exact sampling from this distribution. This demonstrates its applicability for privately answering counting queries, or more generally, low-sensitivity integer-valued queries.

# Contents

---

[*]IBM Research – Almaden. . . . . . . . . . . . . . . . . . . . . . . . . . . . . . . . . . . . . . . . . . . . . . . . . . clement.canonne@ibm.com
[†]University of Waterloo. Supported by a University of Waterloo startup grant. . . . . . . . . . . . . . g@csail.mit.edu
[‡]Work done while at IBM Research – Almaden. Now at Google. . . . . . . . . . . . . . . dgauss@thomas-steinke.net

[0]Alphabetical authorship. Published at NeurIPS 2020. Preprint: https://arxiv.org/abs/2004.00010

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

**Privacy.** The discrete Gaussian enjoys privacy guarantees which are almost identical to those of the continuous Gaussian. More precisely, in Theorem 4, we show that adding noise drawn from $\mathcal{N}_{\mathbb{Z}}\left(0, 1/\varepsilon^2\right)$ to an integer-valued sensitivity-1 query (e.g., a counting query) provides $\frac{1}{2}\varepsilon^2$-concentrated differential privacy. This is the same guarantee attained by adding a draw from $\mathcal{N}(0, 1/\varepsilon^2)$. Furthermore, in Theorem 7, we provide tight bounds on the discrete Gaussian's approximate differential privacy guarantees. For large scales $\sigma$, the discrete and continuous Gaussian have virtually the same privacy guarantee, although for smaller $\sigma$, the effects of discretization result in one or the other having marginally stronger privacy (depending on the parameters). Our results on privacy are presented in Section 2.

**Utility.** The discrete Gaussian attains the same or slightly better accuracy as the analogous continuous Gaussian. Specifically, Corollary 17 shows that the variance of $\mathcal{N}_{\mathbb{Z}}\left(0, \sigma^2\right)$ is at most $\sigma^2$, and that it also satisfies sub-Gaussian tail bounds comparable to $\mathcal{N}(0, \sigma^2)$. We show numerically that the discrete Gaussian is better than rounding the continuous Gaussian to an integral value. Our results on utility are provided in Section 3.

**Sampling.** We can practically sample a discrete Gaussian on a finite computer. We present a simple and efficient exact sampling procedure that only requires access to uniformly random bits and does not involve any real-arithmetic operations or non-trivial function evaluations (Algorithm 3). As there are previous methods (see, e.g., Karney's algorithm [Kar16], which was an inspiration for

our work, and the more recent work of Du, Fan, and Wei [DFW20]), we do not consider this to be one of our primary contributions. Nonetheless, we include these results as we consider our methods to be simpler, and in order to make our paper self-contained; we also provide open source code implementing our algorithm [Dga]. Our results on how to sample are provided in Section 5.

We provide a thorough comparison between the discrete Gaussian and the discrete Laplace distribution in Section 4. This includes statements of the privacy and utility guarantees for the discrete Laplace, and discussing its performance under composition in depth.

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

In this section, we prove our main result on concentrated differential privacy (CDP), showing that the discrete Gaussian provides the same CDP guarantees as the continuous one.

**Theorem 4** (Discrete Gaussian Satisfies Concentrated Differential Privacy). *Let $\Delta, \varepsilon > 0$. Let $q\colon \mathcal{X}^n \to \mathbb{Z}$ satisfy $|q(x) - q(x')| \le \Delta$ for all $x, x' \in \mathcal{X}^n$ differing on a single entry. Define a randomized algorithm $M\colon \mathcal{X}^n \to \mathbb{Z}$ by $M(x) = q(x) + Y$ where $Y \leftarrow \mathcal{N}_\mathbb{Z}\left(0, \Delta^2/\varepsilon^2\right)$. Then $M$ satisfies $\frac{1}{2}\varepsilon^2$-concentrated differential privacy.*

Theorem 4 follows from Proposition 5 and Definition 3 .

**Proposition 5.** *Let $\sigma, \alpha \in \mathbb{R}$ with $\sigma > 0$ and $\alpha \ge 1$. Let $\mu, \nu \in \mathbb{Z}$. Then*

$$\mathrm{D}_\alpha\left(\mathcal{N}_\mathbb{Z}\left(\mu, \sigma^2\right)\|\mathcal{N}_\mathbb{Z}\left(\nu, \sigma^2\right)\right) \le \frac{(\mu - \nu)^2}{2\sigma^2} \cdot \alpha. \tag{2}$$

*Furthermore, this inequality is an equality whenever $\alpha \cdot (\mu - \nu)$ is an integer.*

It is worth noting that the continuous Gaussian satisfies the same concentrated differential privacy bound, with equality for all Rényi divergence parameters: $\mathrm{D}_\alpha\left(\mathcal{N}(\mu, \sigma^2)\|\mathcal{N}(\nu, \sigma^2)\right) = \frac{(\mu-\nu)^2}{2\sigma^2} \cdot \alpha$ for all $\alpha, \mu, \nu, \sigma \in \mathbb{R}$ with $\sigma > 0$. Thus we see that the privacy guarantee of the discrete Gaussian is essentially identical to that of the continuous Gaussian with the same parameters. To prove Proposition 5, we use the following well-known (e.g., [Reg09]) technical lemma.

**Lemma 6.** *Let $\mu, \sigma \in \mathbb{R}$ with $\sigma > 0$. Then*

$$\sum_{x \in \mathbb{Z}} e^{-(x-\mu)^2/2\sigma^2} \leq \sum_{x \in \mathbb{Z}} e^{-x^2/2\sigma^2}. \tag{3}$$

*Proof.* Let $f \colon \mathbb{R} \to \mathbb{R}$ be defined by $f(x) = e^{-x^2/2\sigma^2}$. Define its Fourier transform $\hat{f} \colon \mathbb{R} \to \mathbb{R}$ by

$$\hat{f}(y) = \int_{\mathbb{R}} f(x) e^{-2\pi\sqrt{-1}xy} \mathrm{d}x = \sqrt{2\pi\sigma^2} \cdot e^{-2\pi^2\sigma^2 y^2}.$$

By the Poisson summation formula [Poi; Wei], for every $t \in \mathbb{R}$, we have

$$\sum_{x \in \mathbb{Z}} f(x + t) = \sum_{y \in \mathbb{Z}} \hat{f}(y) \cdot e^{2\pi\sqrt{-1}yt}.$$

(This is the Fourier series representation of the 1-periodic function $g \colon \mathbb{R} \to \mathbb{R}$ given by $g(t) = \sum_{x \in \mathbb{Z}} e^{-(x+t)^2/2\sigma^2}$.) In particular, $f(x) > 0$ and $\hat{f}(x) > 0$ for all $x \in \mathbb{R}$. From this and the triangle inequality, we get

$$\sum_{x \in \mathbb{Z}} e^{-(x-\mu)/2\sigma^2} = \sum_{x \in \mathbb{Z}} f(x - \mu) = \left| \sum_{x \in \mathbb{Z}} f(x - \mu) \right| = \left| \sum_{y \in \mathbb{Z}} \hat{f}(y) e^{-2\pi\sqrt{-1}y\mu} \right|$$

$$\leq \sum_{y \in \mathbb{Z}} \left| \hat{f}(y) e^{-2\pi\sqrt{-1}y\mu} \right| = \sum_{y \in \mathbb{Z}} \hat{f}(y) = \sum_{x \in \mathbb{Z}} f(x) = \sum_{x \in \mathbb{Z}} e^{-x^2/2\sigma^2},$$

proving the lemma. $\qquad\square$

*Proof of Proposition 5.* Without loss of generality, $\nu = 0$ and $\alpha > 1$. Recalling that $\mu \in \mathbb{Z}$, we have

$$e^{(\alpha-1)\mathrm{D}_\alpha\left(\mathcal{N}_{\mathbb{Z}}\left(\mu,\sigma^2\right)\|\mathcal{N}_{\mathbb{Z}}\left(0,\sigma^2\right)\right)} = \sum_{x \in \mathbb{Z}} \mathop{\mathbb{P}}_{X \leftarrow \mathcal{N}_{\mathbb{Z}}(\mu,\sigma^2)} [X = x]^\alpha \cdot \mathop{\mathbb{P}}_{X \leftarrow \mathcal{N}_{\mathbb{Z}}(0,\sigma^2)} [X = x]^{1-\alpha}$$

$$= \sum_{x \in \mathbb{Z}} \left( \frac{e^{-(x-\mu)^2/2\sigma^2}}{\sum_{y \in \mathbb{Z}} e^{-(y-\mu)^2/2\sigma^2}} \right)^\alpha \cdot \left( \frac{e^{-x^2/2\sigma^2}}{\sum_{y \in \mathbb{Z}} e^{-y^2/2\sigma^2}} \right)^{1-\alpha}$$

$$= \frac{\sum_{x \in \mathbb{Z}} \exp\left( \frac{-x^2 + 2\alpha\mu x - \alpha\mu^2}{2\sigma^2} \right)}{\sum_{y \in \mathbb{Z}} e^{-y^2/2\sigma^2}}$$

$$= e^{\alpha(\alpha-1)\mu^2/2\sigma^2} \cdot \frac{\sum_{x \in \mathbb{Z}} e^{-(x-\alpha\mu)^2/2\sigma^2}}{\sum_{y \in \mathbb{Z}} e^{-y^2/2\sigma^2}}$$

$$\leq e^{\alpha(\alpha-1)\mu^2/2\sigma^2},$$

where the final inequality follows from Lemma 6. The inequality is an equality when $\alpha\mu \in \mathbb{Z}$. $\quad\square$

We remark that, like its continuous counterpart, the discrete Gaussian can also be analysed in the setting where the scale parameter $\sigma^2$ is data dependent [BDRS18]. This arises in the application of smooth sensitivity [NRS07; BS19].

## 2.2 Approximate Differential Privacy

In this section, we prove our main result on approximate differential privacy; namely, a tight bound on the privacy parameters achieved by the discrete Gaussian.

**Theorem 7** (Discrete Gaussian Satisfies Approximate Differential Privacy). *Let $\Delta, \sigma, \varepsilon > 0$. Let $q \colon \mathcal{X}^n \to \mathbb{Z}$ satisfy $|q(x) - q(x')| \leq \Delta$ for all $x, x' \in \mathcal{X}^n$ differing on a single entry. Define a randomized algorithm $M \colon \mathcal{X}^n \to \mathbb{Z}$ by $M(x) = q(x) + Y$ where $Y \leftarrow \mathcal{N}_{\mathbb{Z}}\left(0, \sigma^2\right)$. Then $M$ satisfies $(\varepsilon, \delta)$-differential privacy for*

$$\delta = \mathop{\mathbb{P}}_{Y \leftarrow \mathcal{N}_{\mathbb{Z}}(0,\sigma^2)}\left[Y > \frac{\varepsilon\sigma^2}{\Delta} - \frac{\Delta}{2}\right] - e^{\varepsilon} \cdot \mathop{\mathbb{P}}_{Y \leftarrow \mathcal{N}_{\mathbb{Z}}(0,\sigma^2)}\left[Y > \frac{\varepsilon\sigma^2}{\Delta} + \frac{\Delta}{2}\right]. \tag{4}$$

*Furthermore, this is the smallest possible value of $\delta$ for which this is true.*

This privacy guarantee matches that of the continuous Gaussian: If we replace all occurrences of the discrete Gaussian with the continuous Gaussian above, then the same result holds [BW18, Thm. 8]. Empirically, these guarantees are very close.

We also provide some analytic upper bounds (proofs can be found at the end of this section): First, for $\Delta = 1$ we have $\delta \leq e^{-\lfloor \varepsilon\sigma^2 \rceil^2 / 2\sigma^2} / \sqrt{2\pi\sigma^2}$, where $\lfloor \cdot \rceil$ denotes rounding to the nearest integer. Furthermore, if $\varepsilon > \frac{\Delta^2}{2\sigma^2}$ and $\frac{\varepsilon\sigma^2}{\Delta} \pm \frac{\Delta}{2} \notin \mathbb{N}$, then

$$\delta \leq \mathop{\mathbb{P}}_{X \leftarrow \mathcal{N}(0,\sigma^2)}\left[X > \left\lfloor \frac{\varepsilon\sigma^2}{\Delta} - \frac{\Delta}{2} \right\rfloor\right] - \left(1 - \frac{1}{\sqrt{2\pi\sigma^2} + 1}\right) e^{\varepsilon} \mathop{\mathbb{P}}_{X \leftarrow \mathcal{N}(0,\sigma^2)}\left[X > \left\lfloor \frac{\varepsilon\sigma^2}{\Delta} + \frac{\Delta}{2} \right\rfloor\right]. \tag{5}$$

In Figure 1, we empirically compare the optimal $\delta$ (given by Theorem 7) to the bound attained by the corresponding continuous Gaussian, as well as this analytic upper bound (5), the standard upper bound entailed by concentrated differential privacy, and an improved upper bound via concentrated differential privacy (Corollary 13). We see that the upper bounds are reasonably tight. The discrete and continuous Gaussian attain almost identical guarantees for large $\sigma$, but the discretization creates a small difference that becomes apparent for small $\sigma$.

Figure 1: Comparison of approximate $(\varepsilon, \delta)$-differential privacy guarantees ($\delta$ as a function of $\varepsilon$).

To prove Theorem 7, we introduce the privacy loss random variable [DRV10; DR16; BS16] and relate it to approximate differential privacy.[4]

**Definition 8** (Privacy Loss Random Variable). *Let $M \colon \mathcal{X}^n \to \mathbb{Y}$ be a randomized algorithm. Let $x, x' \in \mathcal{X}^n$ be neighbouring inputs. Define $f \colon \mathcal{Y} \to \mathbb{R}$ by $f(y) = \log\left(\frac{\mathbb{P}[M(x)=y]}{\mathbb{P}[M(x')=y]}\right)$. (More formally, $f$ is the logarithm of the Radon-Nikodym derivative of the distribution of $M(x)$ with respect to the distribution of $M(x')$. If the distribution of $M(x)$ is not absolutely continuous with respect to $M(x')$, then the privacy loss random variable is undefined.) Let $Z = f(M(x))$. That is, $Z \in \mathbb{R}$ is the random variable generated by applying the function $f$ to the output of $M(x)$. (The randomness of $Z$ comes entirely from the algorithm $M$.) Then $Z$ is called the* privacy loss random variable *and is denoted $Z \leftarrow \mathsf{PrivLoss}\left(M(x) \| M(x')\right)$.*

Concentrated differential privacy can be formulated in terms of the moment generating function of the privacy loss [BS16]. Specifically, for any $M \colon \mathcal{X}^n \to \mathbb{Y}$, any $x, x' \in \mathcal{X}^n$, and any $\alpha \in (1, \infty)$, we have

$$\mathrm{D}_\alpha\left(M(x) \| M(x')\right) = \frac{1}{\alpha - 1} \log\left(\underset{Z \leftarrow \mathsf{PrivLoss}(M(x)\|M(x'))}{\mathbb{E}}\left[e^{(\alpha-1)Z}\right]\right) \tag{6}$$

Approximate differential privacy can also be characterized via the privacy loss as follows. This characterization is implicit in the work of Bun and Steinke [BS16, Lemma B.2] and is explicit in the work of Meiser and Mohammadi [MM18, Lemma 1] (see also [Goo20a, Observation 2] and references therein).

**Lemma 9.** *Let $\varepsilon, \delta \geq 0$. Let $M: \mathcal{X}^n \to \mathcal{Y}$ be a randomized algorithm. Then $M$ satisfies $(\varepsilon, \delta)$-differential privacy if and only if*

$$\delta \geq \mathop{\mathbb{E}}_{Z \leftarrow \mathsf{PrivLoss}(M(x)\|M(x'))} \left[ \max\{0, 1 - e^{\varepsilon - Z}\} \right] \tag{7}$$

$$= \mathop{\mathbb{P}}_{Z \leftarrow \mathsf{PrivLoss}(M(x)\|M(x'))} [Z > \varepsilon] - e^{\varepsilon} \cdot \mathop{\mathbb{P}}_{Z' \leftarrow \mathsf{PrivLoss}(M(x')\|M(x))} \left[ -Z' > \varepsilon \right] \tag{8}$$

$$= \int_{\varepsilon}^{\infty} e^{\varepsilon - z} \mathop{\mathbb{P}}_{Z \leftarrow \mathsf{PrivLoss}(M(x)\|M(x'))} [Z > z] \mathrm{d}z \tag{9}$$

*for all $x, x' \in \mathcal{X}^n$ differing on a single element.*

Observe that, by Markov's inequality, for all $\alpha > 1$, it suffices to set

$$\delta = \mathop{\mathbb{P}}_{Z \leftarrow \mathsf{PrivLoss}(M(x)\|M(x'))} [Z > \varepsilon]$$

$$\leq e^{-(\alpha-1)\varepsilon} \cdot \mathop{\mathbb{E}}_{Z \leftarrow \mathsf{PrivLoss}(M(x)\|M(x'))} \left[ e^{(\alpha-1)Z} \right]$$

$$= e^{(\alpha-1)(\mathrm{D}_\alpha(M(x)\|M(x'))-\varepsilon)}. \tag{10}$$

This is the usual expression that is used to convert bounds on the privacy loss or Rényi divergence into approximate differential privacy. Lemma 9 and Proposition 12 represent an improvement on this.

*Proof.* Fix neighbouring inputs $x, x' \in \mathcal{X}^n$. Let $f: \mathcal{Y} \to \mathbb{R}$ be as in Definition 8. For notational simplicity, let $Y = M(x)$ and $Y' = M(x')$ and $Z = f(Y)$ and $Z' = -f(Y')$. This is equivalent to $Z \leftarrow \mathsf{PrivLoss}(M(x)\|M(x'))$ and $Z' \leftarrow \mathsf{PrivLoss}(M(x')\|M(x))$. Our first goal is to prove that

$$\sup_{E \subset \mathcal{Y}} \mathbb{P}[Y \in E] - e^{\varepsilon} \mathbb{P}[Y' \in E] = \mathbb{E}\left[ \max\{0, 1 - e^{\varepsilon - Z}\} \right].$$

For any $E \subset \mathcal{Y}$, we have

$$\mathbb{P}[Y' \in E] = \mathbb{E}\left[ \mathbb{I}[Y' \in E] \right] = \mathbb{E}\left[ \mathbb{I}[Y \in E] \cdot e^{-f(Y)} \right].$$

This is because $e^{-f(y)} = \frac{\mathbb{P}[Y=y]}{\mathbb{P}[Y'=y]}$.[5]
Thus, for all $E \subset \mathcal{Y}$, we have

$$\mathbb{P}[Y \in E] - e^{\varepsilon} \mathbb{P}[Y' \in E] = \mathbb{E}\left[ \mathbb{I}[Y \in E] \cdot (1 - e^{\varepsilon - f(Y)}) \right].$$

Now it is easy to identify the worst event as $E = \{y \in \mathcal{Y} : 1 - e^{\varepsilon - f(y)} > 0\}$. Thus

$$\sup_{E \subset \mathcal{Y}} \mathbb{P}[Y \in E] - e^{\varepsilon} \mathbb{P}[Y' \in E] = \mathbb{E}\left[ \mathbb{I}[1 - e^{\varepsilon - f(Y)} > 0] \cdot (1 - e^{\varepsilon - f(Y)}) \right] = \mathbb{E}\left[ \max\{0, 1 - e^{\varepsilon - Z}\} \right].$$

Alternatively, since the worst event is equivalently $E = \{y \in \mathcal{Y} : f(y) > \varepsilon\}$, we have

$$\sup_{E \subset \mathcal{Y}} \mathbb{P}[Y \in E] - e^{\varepsilon} \mathbb{P}[Y' \in E] = \mathbb{P}[f(Y) > \varepsilon] - e^{\varepsilon} \mathbb{P}[f(Y') > \varepsilon] = \mathbb{P}[Z > \varepsilon] - e^{\varepsilon} \mathbb{P}[-Z' > \varepsilon].$$

It only remains to show that

$$\mathbb{E}\left[\max\{0, 1 - e^{\varepsilon - Z}\}\right] = \int_{\varepsilon}^{\infty} e^{\varepsilon - z}\mathbb{P}\left[Z > z\right]\mathrm{d}z.$$

This follows from integration by parts: Let $u(z) = \mathbb{P}\left[Z > z\right]$ and $v(z) = 1 - e^{\varepsilon - z}$ and $w(z) = u(z) \cdot v(z)$. Then

$$\mathbb{E}\left[\max\{0, 1 - e^{\varepsilon - Z}\}\right] = \int_{\varepsilon}^{\infty} v(z) \cdot u'(z)\mathrm{d}z = \int_{\varepsilon}^{\infty} \left(w'(z) - v'(z) \cdot u(z)\right)\mathrm{d}z$$

$$= \lim_{z \to \infty} w(z) - w(\varepsilon) + \int_{\varepsilon}^{\infty} e^{\varepsilon - z}\mathbb{P}\left[Z > z\right]\mathrm{d}z.$$

Now $w(\varepsilon) = u(\varepsilon) \cdot (1 - e^{\varepsilon - \varepsilon}) = 0$ and $0 \le \lim_{z \to \infty} w(z) \le \lim_{z \to \infty} \mathbb{P}\left[Z > z\right] = 0$, as required. $\qquad\square$

*Proof of Theorem 7.* We will use Lemma 9, Equation 8. Thus our main task is to determine the distribution of the privacy loss random variable.

Fix neighbouring inputs $x, x' \in \mathcal{X}^n$. Without loss of generality, we may assume that $q(x) = 0$ and $q(x') > 0$. Now we have $q(x') \le \Delta$. Let $f : \mathcal{Y} \to \mathbb{R}$ be as in Definition 8. For notational simplicity, let $Y = M(x) \sim \mathcal{N}_{\mathbb{Z}}\left(0, \sigma^2\right)$ and $Y' = M(x') \sim \mathcal{N}_{\mathbb{Z}}\left(q(x'), \sigma^2\right)$ and $Z = f(Y)$ and $Z' = -f(Y')$. This is equivalent to $Z \leftarrow \mathsf{PrivLoss}\left(M(x)\|M(x')\right)$ and $Z' \leftarrow \mathsf{PrivLoss}\left(M(x')\|M(x)\right)$. We must show that

$$\mathbb{P}\left[Z > \varepsilon\right] - e^{\varepsilon}\mathbb{P}\left[-Z' > \varepsilon\right] \le \Pr_{Y \leftarrow \mathcal{N}_{\mathbb{Z}}(0,\sigma^2)}\left[Y > \frac{\varepsilon\sigma^2}{\Delta} - \frac{\Delta}{2}\right] - e^{\varepsilon} \cdot \Pr_{Y \leftarrow \mathcal{N}_{\mathbb{Z}}(0,\sigma^2)}\left[Y > \frac{\varepsilon\sigma^2}{\Delta} + \frac{\Delta}{2}\right]$$

For $y \in \mathcal{Y}$, we have

$$f(y) = \log\left(\frac{\mathbb{P}\left[Y = y\right]}{\mathbb{P}\left[Y' = y\right]}\right) = \log\left(\frac{e^{-(y-q(x))^2/2\sigma^2}}{e^{-(y-q(x'))^2/2\sigma^2}}\right) = \frac{(y - q(x'))^2 - y^2}{2\sigma^2} = \frac{q(x') \cdot (q(x') - 2y)}{2\sigma^2}.$$

Thus, for all $y \in \mathcal{Y}$,

$$f(y) > \varepsilon \iff -y > \frac{\sigma^2\varepsilon}{q(x')} - \frac{q(x')}{2}.$$

We note that $Y$ and $-Y$ and $Y' - q(x')$ and $q(x') - Y'$ all have the same distribution. Hence

$$\mathbb{P}\left[Z > \varepsilon\right] = \mathbb{P}\left[f(Y) > \varepsilon\right] = \mathbb{P}\left[-Y > \frac{\sigma^2\varepsilon}{q(x')} - \frac{q(x')}{2}\right] = \mathbb{P}\left[Y > \frac{\sigma^2\varepsilon}{q(x')} - \frac{q(x')}{2}\right]$$

and

$$\mathbb{P}\left[-Z' > \varepsilon\right] = \mathbb{P}\left[-Y' > \frac{\sigma^2\varepsilon}{q(x')} - \frac{q(x')}{2}\right] = \mathbb{P}\left[Y - q(x') > \frac{\sigma^2\varepsilon}{q(x')} - \frac{q(x')}{2}\right] = \mathbb{P}\left[Y > \frac{\sigma^2\varepsilon}{q(x')} + \frac{q(x')}{2}\right].$$

$\qquad\square$

**Proofs of the analytical bounds.** We now provide the proofs of the two aforementioned analytical bounds for $(\varepsilon, \delta)$-differential privacy our theorem readily implies.

**Lemma 10.** *In the setting of Theorem 7, for $\Delta = 1$, we have $\delta \leq e^{-\lfloor \varepsilon \sigma^2 \rceil^2 / 2\sigma^2} / \sqrt{2\pi\sigma^2}$, where $\lfloor \cdot \rceil$ denotes rounding to the nearest integer. More generally, $\delta \leq \sum_{k=\lceil \varepsilon\sigma^2/\Delta - \Delta/2 \rceil}^{\lceil \varepsilon\sigma^2/\Delta + \Delta/2 \rceil} \frac{e^{-k^2/2\sigma^2}}{\sqrt{2\pi\sigma^2}}$*

*Proof.* By Theorem 7, we have

$$
\begin{aligned}
\delta &= \mathbb{P}_{Y \leftarrow \mathcal{N}_{\mathbb{Z}}(0,\sigma^2)}\left[ Y > \frac{\varepsilon\sigma^2}{\Delta} - \frac{\Delta}{2} \right] - e^{\varepsilon} \cdot \mathbb{P}_{Y \leftarrow \mathcal{N}_{\mathbb{Z}}(0,\sigma^2)}\left[ Y > \frac{\varepsilon\sigma^2}{\Delta} + \frac{\Delta}{2} \right] \\
&\leq \mathbb{P}_{Y \leftarrow \mathcal{N}_{\mathbb{Z}}(0,\sigma^2)}\left[ Y > \frac{\varepsilon\sigma^2}{\Delta} - \frac{\Delta}{2} \right] - \mathbb{P}_{Y \leftarrow \mathcal{N}_{\mathbb{Z}}(0,\sigma^2)}\left[ Y > \frac{\varepsilon\sigma^2}{\Delta} + \frac{\Delta}{2} \right] \\
&= \sum_{k=\lceil \varepsilon\sigma^2/\Delta - \Delta/2 \rceil}^{\lceil \varepsilon\sigma^2/\Delta + \Delta/2 \rceil} \mathbb{P}_{Y \leftarrow \mathcal{N}_{\mathbb{Z}}(0,\sigma^2)}[Y = k]
\end{aligned}
$$

and the result now follows from the bound on the normalization constant from Fact 19. $\qquad \square$

**Lemma 11.** *In the setting of Theorem 7, if $\varepsilon > \frac{\Delta^2}{2\sigma^2}$ and $\frac{\varepsilon\sigma^2}{\Delta} + \frac{\Delta}{2}, \frac{\varepsilon\sigma^2}{\Delta} - \frac{\Delta}{2} \notin \mathbb{N}$ then*

$$
\delta \leq \mathbb{P}_{X \leftarrow \mathcal{N}(0,\sigma^2)}\left[ X > \left\lfloor \frac{\varepsilon\sigma^2}{\Delta} - \frac{\Delta}{2} \right\rfloor \right] - \left( 1 - \frac{1}{\sqrt{2\pi\sigma^2} + 1} \right) e^{\varepsilon} \mathbb{P}_{X \leftarrow \mathcal{N}(0,\sigma^2)}\left[ X > \left\lfloor \frac{\varepsilon\sigma^2}{\Delta} + \frac{\Delta}{2} \right\rfloor \right]. \quad (11)
$$

*Proof.* Let $\varepsilon > \frac{\Delta^2}{2\sigma^2}$ be such that $\frac{\varepsilon\sigma^2}{\Delta} + \frac{\Delta}{2} \notin \mathbb{N}$, and set $M(\varepsilon, \sigma) := \lceil \varepsilon\sigma^2/\Delta - \Delta/2 \rceil$ and $m(\varepsilon, \sigma) := \lfloor \varepsilon\sigma^2/\Delta + \Delta/2 \rfloor$. Then, by Proposition 25,

$$
\begin{aligned}
\mathbb{P}_{X \leftarrow \mathcal{N}_{\mathbb{Z}}(0,\sigma^2)}\left[ X > \frac{\varepsilon\sigma^2}{\Delta} - \frac{\Delta}{2} \right] &= \mathbb{P}_{X \leftarrow \mathcal{N}_{\mathbb{Z}}(0,\sigma^2)}[X \geq M(\varepsilon, \sigma)] \leq \mathbb{P}_{X \leftarrow \mathcal{N}(0,\sigma^2)}[X \geq M(\varepsilon, \sigma) - 1] \\
&\leq \mathbb{P}_{X \leftarrow \mathcal{N}(0,\sigma^2)}\left[ X \geq \left\lfloor \frac{\varepsilon\sigma^2}{\Delta} - \frac{\Delta}{2} \right\rfloor \right]
\end{aligned}
$$

Conversely, by a comparison series-integral, we can easily show that, for any integer $m$,

$$
\sum_{n=m}^{\infty} e^{-n^2/(2\sigma^2)} = \sum_{n=m}^{\infty} \int_{n}^{n+1} e^{-n^2/(2\sigma^2)} \, \mathrm{d}x \geq \int_{m}^{\infty} e^{-x^2/(2\sigma^2)} \, \mathrm{d}x = \sqrt{2\pi\sigma^2} \mathbb{P}_{X \leftarrow \mathcal{N}(0,\sigma^2)}[X \geq m]
$$

which, combined with Fact 19 on the normalization constant of the discrete Gaussian, yields

$$
\mathbb{P}_{X \leftarrow \mathcal{N}_{\mathbb{Z}}(0,\sigma^2)}\left[ X > \frac{\varepsilon\sigma^2}{\Delta} + \frac{\Delta}{2} \right] = \mathbb{P}_{X \leftarrow \mathcal{N}_{\mathbb{Z}}(0,\sigma^2)}[X \geq m(\varepsilon, \sigma)] \geq \frac{1}{1 + \frac{1}{\sqrt{2\pi\sigma^2}}} \mathbb{P}_{X \leftarrow \mathcal{N}(0,\sigma^2)}[X \geq m(\varepsilon, \sigma)]
$$

The result then follows from Theorem 7. $\qquad \square$

## 2.3 Converting Concentrated Differential Privacy to Approximate Differential Privacy

We have stated guarantees for both concentrated differential privacy (Theorem 4) and approximate differential privacy (Theorem 7). Now we show how to convert from the former to the latter (Corollary 13). This is particularly useful if the discrete Gaussian is being used repeatedly and we want to provide a privacy guarantee for the composition – concentrated differential privacy has cleaner composition guarantees than approximate differential privacy. We include this result for completeness; this result was recently proved independently [ALCKS20, Lem. 1, Eq. 20].

We start with a conversion from Rényi differential privacy to approximate differential privacy.

**Proposition 12.** *Let* $M \colon \mathcal{X}^n \to \mathcal{Y}$ *be a randomized algorithm. Let* $\alpha \in (1, \infty)$ *and* $\varepsilon \geq 0$. *Suppose* $\mathrm{D}_\alpha\left(M(x) \| M(x')\right) \leq \tau$ *for all* $x, x' \in \mathcal{X}^n$ *differing in a single entry.[6] Then* $M$ *is* $(\varepsilon, \delta)$*-differentially private for*

$$\delta = \frac{e^{(\alpha-1)(\tau-\varepsilon)}}{\alpha - 1} \cdot \left(1 - \frac{1}{\alpha}\right)^\alpha \leq \frac{e^{(\alpha-1)(\tau-\varepsilon)-1}}{\alpha - 1}. \tag{12}$$

In contrast, the standard bound [DRV10; DR16; BS16; Mir17] is $\delta \leq e^{(\alpha-1)(\tau-\varepsilon)}$. Note that $\frac{e^{(\alpha-1)(\tau-\varepsilon)}}{\alpha-1} \cdot \left(1 - \frac{1}{\alpha}\right)^\alpha = \frac{e^{(\alpha-1)(\tau-\varepsilon)}}{\alpha} \cdot \left(1 - \frac{1}{\alpha}\right)^{\alpha-1}$. Thus Proposition 12 is strictly better than the standard bound for $\alpha > 1$. Equation 12 can be rearranged to

$$\varepsilon = \tau + \frac{\log(1/\delta) + (\alpha-1)\log(1 - 1/\alpha) - \log(\alpha)}{\alpha - 1}. \tag{13}$$

*Proof.* Fix neighbouring $x, x' \in \mathcal{X}^n$ and let $Z \leftarrow \mathsf{PrivLoss}\left(M(x) \| M(x')\right)$. We have

$$\mathbb{E}\left[e^{(\alpha-1)Z}\right] = e^{(\alpha-1)\mathrm{D}_\alpha(M(x)\|M(x'))} \leq e^{(\alpha-1)\tau}.$$

By Lemma 9, our goal is to prove that $\delta \geq \mathbb{E}\left[\max\{0, 1 - e^{\varepsilon-Z}\}\right]$. Our approach is to pick $c > 0$ such that $\max\{0, 1 - e^{\varepsilon-z}\} \leq c \cdot e^{(\alpha-1)z}$ for all $z \in \mathbb{R}$. Then

$$\mathbb{E}\left[\max\{0, 1 - e^{\varepsilon-Z}\}\right] \leq \mathbb{E}\left[c \cdot e^{(\alpha-1)Z}\right] \leq c \cdot e^{(\alpha-1)\tau}.$$

We identify the smallest possible value of $c$:

$$c = \sup_{z \in \mathbb{R}} \frac{\max\{0, 1 - e^{\varepsilon-z}\}}{e^{(\alpha-1)z}} = \sup_{z \in \mathbb{R}} e^{z - \alpha \cdot z} - e^{\varepsilon - \alpha \cdot z} = \sup_{z \in \mathbb{R}} f(z),$$

where $f(z) = e^{z - \alpha \cdot z} - e^{\varepsilon - \alpha \cdot z}$. We have

$$f'(z) = e^{z-\alpha z}(1 - \alpha) - e^{\varepsilon - \alpha z}(-\alpha) = e^{-\alpha z}(\alpha e^\varepsilon - (\alpha - 1)e^z).$$

Clearly $f'(z) = 0 \iff e^z = \frac{\alpha}{\alpha-1}e^\varepsilon \iff z = \varepsilon - \log(1 - 1/\alpha)$. Thus

$$c = f(\varepsilon - \log(1 - 1/\alpha))$$

$$= \left(\frac{\alpha}{\alpha-1}e^\varepsilon\right)^{1-\alpha} - e^\varepsilon \cdot \left(\frac{\alpha}{\alpha-1}e^\varepsilon\right)^{-\alpha}$$

$$= \left(\frac{\alpha}{\alpha-1}e^\varepsilon - e^\varepsilon\right) \cdot \left(\frac{\alpha-1}{\alpha} \cdot e^{-\varepsilon}\right)^\alpha$$

$$= \frac{e^\varepsilon}{\alpha-1} \cdot \left(1 - \frac{1}{\alpha}\right)^\alpha \cdot e^{-\alpha\varepsilon}.$$

Thus

$$\mathbb{E}\left[\max\{0, 1 - e^{\varepsilon - Z}\}\right] \leq \frac{e^{\varepsilon}}{\alpha - 1} \cdot \left(1 - \frac{1}{\alpha}\right)^{\alpha} \cdot e^{-\alpha\varepsilon} \cdot e^{(\alpha-1)\tau} = \frac{e^{(\alpha-1)(\tau-\varepsilon)}}{\alpha - 1} \cdot \left(1 - \frac{1}{\alpha}\right)^{\alpha} = \delta. \quad \square$$

Asoodeh, Liao, Calmon, Kosut, and Sankar [ALCKS20] provide an optimal conversion from Rényi differential privacy to approximate differential privacy – i.e., an optimal version of Proposition 12. Specifically, the optimal bound is

$$\delta = \inf\left\{\hat{\delta} \in [0,1] : \forall p \in (\hat{\delta}, 1) \ \ p^{\alpha}(p - \hat{\delta})^{1-\alpha} + (1-p)^{\alpha}(e^{\varepsilon} - p + \hat{\delta})^{1-\alpha} \leq e^{(\alpha-1)(\tau-\varepsilon)}\right\}. \quad (14)$$

Clearly, the expression in Proposition 12 is simpler than this. Moreover, our expression is numerically stable, whereas the alternative is unstable for small values of $\delta$. We implemented both methods and found that they yield very similar results for the parameter regime of interest, but numerical stability was a significant practical issue.

By taking the infimum over all divergence parameters $\alpha$, Proposition 12 entails the following conversion from concentrated differential privacy to approximate differential privacy.

**Corollary 13.** *Let* $M\colon \mathcal{X}^n \to \mathcal{Y}$ *be a randomized algorithm satisfying $\rho$-concentrated differential privacy. Then $M$ is $(\varepsilon, \delta)$-differentially private for any $\varepsilon \geq 0$ and*

$$\delta = \inf_{\alpha \in (1,\infty)} \frac{e^{(\alpha-1)(\alpha\rho-\varepsilon)}}{\alpha - 1} \cdot \left(1 - \frac{1}{\alpha}\right)^{\alpha} \leq \inf_{\alpha \in (1,\infty)} \frac{e^{(\alpha-1)(\alpha\rho-\varepsilon)-1}}{\alpha - 1}. \quad (15)$$

Corollary 13 should be contrasted with the standard bound [DRV10; DR16; BS16; Mir17] of

$$\delta = \inf_{\alpha \in (1,\infty)} e^{(\alpha-1)(\alpha\rho-\varepsilon)} = e^{-(\varepsilon-\rho)^2/4\rho}, \quad (16)$$

which holds when $\varepsilon \geq \rho > 0$. Bun and Steinke [BS16] prove an intermediate bound of

$$\delta = 2\sqrt{\pi\rho} \cdot e^{\varepsilon} \cdot \mathbb{P}_{X \leftarrow \mathcal{N}(0,1)}\left[X > \frac{\varepsilon + \rho}{\sqrt{2\rho}}\right] \quad (17)$$

**Efficient computation of $\delta$.** For the looser expression in Corollary 13, we can analytically find an optimal $\alpha$. However, we can efficiently compute a tighter numerical bound: The equality in Equation 15 is equivalent to

$$\delta = \inf_{\alpha \in (1,\infty)} e^{g(\alpha)} \qquad \text{where} \qquad g(\alpha) = (\alpha-1)(\alpha\rho - \varepsilon) + (\alpha - 1) \cdot \log(1 - 1/\alpha) - \log(\alpha). \quad (18)$$

We have

$$g'(\alpha) = (2\alpha - 1)\rho - \varepsilon + \log(1 - 1/\alpha) \quad (19)$$

and

$$g''(\alpha) = 2\rho + \frac{1}{\alpha(\alpha - 1)} > 0. \quad (20)$$

Since $g$ is a smooth convex function with[7]

$$g'\left(\frac{\varepsilon + \rho}{2\rho}\right) = 0 + \log\left(\frac{\varepsilon - \rho}{\varepsilon + \rho}\right) < 0 \quad (21)$$

and

$$g'\left(\max\left\{\frac{\varepsilon+\rho+1}{2\rho},2\right\}\right) \geq 1 - \log 2 > 0, \tag{22}$$

it has a unique minimizer $\alpha_* \in \left(\frac{\varepsilon+\rho}{2\rho}, \max\{\frac{\varepsilon+\rho+1}{2\rho},2\}\right)$. We can find the minimizer $\alpha_*$ by conducting a binary search over the interval $\left(\frac{\varepsilon+\rho}{2\rho}, \max\{\frac{\varepsilon+\rho+1}{2\rho},2\}\right)$. That is, we want to find $\alpha_*$ such that $g'(\alpha_*) = 0$; if $\alpha < \alpha_*$, we have $g'(\alpha) < 0$ and, if $\alpha > \alpha_*$, we have $g'(\alpha) > 0$.

## 2.4 Sharp Approximate Differential Privacy Bounds for Multivariate Noise

Next we consider adding independent discrete Gaussians to a multivariate function. We begin with a concentrated differential privacy bound:

**Theorem 14** (Multivariate Discrete Gaussian Satisfies Concentrated Differential Privacy)**.** *Let* $\sigma_1, \cdots, \sigma_d > 0$ *and* $\varepsilon > 0$. *Let* $q \colon \mathcal{X}^n \to \mathbb{Z}^d$ *satisfy* $\sum_{j\in[d]}(q_j(x)-q_j(x'))^2/\sigma_j^2 \leq \varepsilon^2$ *for all* $x, x' \in \mathcal{X}^n$ *differing on a single entry. Define a randomized algorithm* $M \colon \mathcal{X}^n \to \mathbb{Z}^d$ *by* $M(x) = q(x) + Y$ *where* $Y_j \leftarrow \mathcal{N}_{\mathbb{Z}}\left(0, \sigma_j^2\right)$ *independently for all* $j \in [d]$. *Then* $M$ *satisfies* $\frac{1}{2}\varepsilon^2$-*concentrated differential privacy.*

Theorem 14 follows from Proposition 5, composition of concentrated differential privacy, and Definition 3. If $\sigma_1 = \sigma_2 = \cdots = \sigma_d$, then the concentrated differential privacy guarantee depends only on the sensitivity of $q$ in the Euclidean norm; if the $\sigma_j$s are different, then it is a weighted Euclidean norm. Note that we only consider multivariate Gaussians with independent coordinates.

It is possible to obtain an approximate differential privacy guarantee for the multivariate discrete Gaussian from Theorem 14 and Corollary 13. While this bound is reasonably tight, we will now give an exact bound:

**Theorem 15.** *Let* $\sigma_1, \cdots, \sigma_d > 0$. *Let* $Y_j \leftarrow \mathcal{N}_{\mathbb{Z}}\left(0, \sigma_j^2\right)$ *independently for each* $j \in [d]$. *Let* $q \colon \mathcal{X}^n \to \mathbb{Z}^d$. *Define a randomized algorithm* $M \colon \mathcal{X}^n \to \mathbb{Z}^d$ *by* $M(x) = q(x) + Y$. *Let* $\varepsilon, \delta > 0$. *Then* $M$ *is* $(\varepsilon, \delta)$-*differentially private if, and only if, for all* $x, x' \in \mathcal{X}^n$ *differing on a single entry, we have*

$$\delta \geq \mathbb{E}\left[\max\left\{0, 1 - \exp\left(\varepsilon - Z\right)\right\}\right] \tag{23}$$

$$= \mathbb{P}\left[Z > \varepsilon\right] - e^\varepsilon \cdot \mathbb{P}\left[Z < -\varepsilon\right] \tag{24}$$

$$= \int_\varepsilon^\infty e^{\varepsilon-z} \cdot \mathbb{P}\left[Z > z\right] \, \mathrm{d}z, \tag{25}$$

*where*

$$Z := \sum_{j=1}^d \frac{(q(x)_j - q(x')_j)^2 + 2(q(x)_j - q(x')_j) \cdot Y_j}{2\sigma_j^2}. \tag{26}$$

*Proof.* Fix neighbouring $x, x' \in \mathcal{X}^n$. Without loss of generality, we may assume $q(x) = 0$. Following the proof of Theorem 7, we will apply Lemma 9, which requires understanding the privacy loss random variable.

Let $f\colon \mathbb{Z}^d \to \mathbb{R}$ be as in Definition 8. That is,

$$
\begin{aligned}
f(y) &= \log\left(\frac{\mathbb{P}\left[M(x) = y\right]}{\mathbb{P}\left[M(x') = y\right]}\right) \\
&= \sum_{j=1}^{d} \log\left(\frac{\mathbb{P}_{Y_j \leftarrow \mathcal{N}_{\mathbb{Z}}\left(0,\sigma_j^2\right)}\left[q(x)_j + Y_j = y_j\right]}{\mathbb{P}_{Y_j \leftarrow \mathcal{N}_{\mathbb{Z}}\left(0,\sigma_j^2\right)}\left[q(x')_j + Y_j = y_j\right]}\right) \\
&= \sum_{j=1}^{d} \frac{-y_j^2 + (y_j - q(x')_j)^2}{2\sigma_j^2} \\
&= \sum_{j=1}^{d} \frac{q(x')_j^2 - 2y_j \cdot q(x')_j}{2\sigma_j^2}.
\end{aligned}
$$

Then the privacy loss $Z \leftarrow \mathsf{PrivLoss}\left(M(x)\|M(x')\right)$ is given by

$$
Z = f(Y) = \sum_{j=1}^{d} \frac{q(x')_j^2 - 2q(x')_j Y_j}{2\sigma_j^2}.
$$

Substituting this expression into Equations 7 and 9 yields Equations 23 and 25 respectively.

Next we look at $Z' \leftarrow \mathsf{PrivLoss}\left(M(x')\|M(x)\right)$, which is given by

$$
Z' = -f(q(x') + Y) = -\sum_{j=1}^{d} \frac{q(x')_j^2 - 2q(x')_j(Y_j + q(x')_j)}{2\sigma_j^2} = \sum_{j=1}^{d} \frac{q(x')_j^2 + 2q(x')_j Y_j}{2\sigma_j^2}.
$$

Noting that each $Y_j$ has a symmetric distribution, we see that $Z'$ has the same distribution as $Z$. Substituting these expressions into Equation 8 yields Equation 24. $\qquad \square$

Theorem 14 gives three equivalent expressions for the approximate differential privacy guarantee of the multivariate discrete Gaussian. All of these expressions are in terms of the privacy loss random variable $Z \leftarrow \mathsf{PrivLoss}\left(M(x)\|M(x')\right)$. We make some remarks about evaluating these expressions:

1. Direct evaluation of the expressions is often impractical. Computing the distribution of $Z$ entails evaluating an infinite sum. Fortunately the terms decay rapidly, so the sum can be truncated, but this still leaves a number of terms that grows exponentially in the dimensionality $d$. Thus we must find more effective ways to evaluate the expressions.

2. The approach underlying concentrated differential privacy is to consider the moment generating function $e^{(\alpha-1)\mathrm{D}_\alpha(M(x)\|M(x'))} = \mathbb{E}\left[e^{(\alpha-1)Z}\right]$. This provides reasonably tight upper bounds on the approximate differential privacy guarantees. However, this approach is not suitable for numerically computing exact bounds [McC94].

3. Instead of the moment generating function, we consider the characteristic function: Let $\sigma_1, \cdots, \sigma_d > 0$. Let $Y_j \leftarrow \mathcal{N}_{\mathbb{Z}}\left(0, \sigma_j^2\right)$ independently for each $j \in [d]$. Let $\mu = q(x) - q(x') \in \mathbb{Z}^d$

and $Z := \sum_{j=1}^{d} \frac{\mu_j^2 + 2\mu_j \cdot Y_j}{2\sigma_j^2}$. The characteristic function of the discrete Gaussian can be expressed two ways: Let $i = \sqrt{-1}$. For $t \in \mathbb{R}$ and all $j \in [d]$, we have

$$\mathbb{E}\left[e^{itY_j}\right] = \frac{\sum_{y \in \mathbb{Z}} e^{ity - y^2/2\sigma_j^2}}{\sum_{y \in \mathbb{Z}} e^{-y^2/2\sigma_j^2}} \tag{27}$$

$$= \frac{\sum_{u \in \mathbb{Z}} e^{-(t - 2\pi u)^2 \sigma_j^2/2}}{\sum_{u \in \mathbb{Z}} e^{-(2\pi u)^2 \sigma_j^2/2}}. \tag{28}$$

The equivalence of the second expression (28) follows from the Poisson summation formula. When $2\pi^2 \sigma_j^2 > 1/2\sigma_j^2$, then the second expression converges more rapidly; otherwise the first expression converges faster. In either case, accurately evaluating the characteristic function of the discrete Gaussian is easy.

It is then possible to compute the characteristic function of the privacy loss:

$$\mathbb{E}\left[e^{itZ}\right] = \prod_{j}^{d} \left( e^{it \frac{\mu_j^2}{2\sigma_j^2}} \cdot \mathbb{E}\left[e^{it \frac{\mu_j}{\sigma_j^2} Y_j}\right] \right). \tag{29}$$

4. Since the discrete Gaussian is symmetric about 0, its characteristic function is real-valued. Since the discrete Gaussian is supported on the integers, its characteristic function is periodic: $\mathbb{E}\left[e^{i(t+2\pi)Y_j}\right] = \mathbb{E}\left[e^{itY_j}\right]$ for all $t \in \mathbb{R}$ and all $j \in [d]$.

5. Assume there exists some $\gamma > 0$ such that, for all $j \in [d]$, there exists $v \in \mathbb{Z}$ satisfying $1/\sigma_j^2 = \gamma \cdot v$. This assumption holds if $\sigma_j^2$ is rational for all $j \in [d]$.

   Under this assumption the privacy loss is always an integer multiple of $\gamma$ – i.e., $\mathbb{P}\left[Z \in \gamma\mathbb{Z}\right] = 1$. Consequently the characteristic function of the privacy loss is also periodic – i.e., $\mathbb{E}\left[e^{i(t+2\pi/\gamma)Z}\right] = \mathbb{E}\left[e^{itZ}\right]$ for all $t \in \mathbb{R}$.

6. It is possible to compute the probability mass function of the privacy loss from the characteristic function:

$$\forall z \in \gamma\mathbb{Z} \qquad \mathbb{P}\left[Z = z\right] = \frac{\gamma}{2\pi} \int_0^{2\pi/\gamma} e^{-itz} \cdot \mathbb{E}\left[e^{itZ}\right] \mathrm{d}t \tag{30}$$

   This can form the basis of an algorithm for computing the guarantee of Theorem 14: The characteristic function can be easily computed from Equations 27, 28, and 29 and then we numerically integrate it according to Equation 30 to compute the probability distribution of the privacy loss and finally we substitute this into Equation 23.

   The downside of this approach is that (i) it requires numerical integration and (ii) it only gives us the probabilities one at a time. Both of these downsides could make the procedure quite slow.

7. We propose to use the discrete Fourier transform (a.k.a. fast Fourier transform) to avoid these downsides.

Effectively, we will compute the distribution of $Z$ modulo $m\gamma$ for some integer $m$. (For fast computation, $m$ should be a power of two.) Call this modular random variable $Z_m$, so that

$$\mathbb{P}[Z_m = z] = \sum_{k \in \mathbb{Z}} \mathbb{P}[Z = z + m\gamma k].$$

Rather than taking $Z_m$ to be supported on $\{0, \gamma, \cdots, (m-1)\gamma\}$ as is usual, we will take $Z_m$ to be supported on $\{(1 - m/2)\gamma, (2 - m/2)\gamma, \cdots, (m/2 - 1)\gamma, (m/2)\gamma\}$.

We will choose $m$ large enough so that $\mathbb{P}[Z \neq Z_m]$ is sufficiently small.

8. The inverse discrete Fourier transform allows us to compute the probability mass of $Z_m$ from the characteristic function of $Z$ (which is identical to the characteristic function of $Z_m$ at the points of interest):

$$\mathbb{P}[Z_m = z] = \frac{1}{m} \sum_{k=0}^{m-1} e^{-i2\pi zk/m\gamma} \cdot \mathbb{E}\left[e^{i2\pi kZ/m\gamma}\right] \tag{31}$$

The fast Fourier transform uses a divide-and-conquer approach to allow us to compute the entire distribution of $Z_m$ in nearly linear time from the values $\mathbb{E}\left[e^{i2\pi kZ/m\gamma}\right]$ for $k = 0 \cdots m-1$. These values can be easily computed using Equations 28 and 29.

9. Now we can compute an upper bound on the approximate differential privacy guarantee (23) using the inequality

$$\delta = \mathbb{E}\left[\max\{0, 1 - e^{\varepsilon - Z}\}\right] \leq \mathbb{E}\left[\max\{0, 1 - e^{\varepsilon - Z_m}\}\right] + \mathbb{P}[Z > Z_m]. \tag{32}$$

10. It only remains to bound $\mathbb{P}[Z > Z_m]$. For $\alpha > 1$, we have

$$\mathbb{P}[Z > Z_m] = \mathbb{P}[Z > (m/2)\gamma] \leq \mathbb{E}\left[e^{(\alpha-1)(Z-(m/2)\gamma)}\right] = e^{(\alpha-1)(D_\alpha(M(x)\|M(x'))-m\gamma/2)}. \tag{33}$$

From the concentrated differential privacy analysis, we have $D_\alpha(M(x)\|M(x')) \leq \alpha \cdot \sum_j^d \frac{\mu_j^2}{2\sigma_j^2}$ for all $\alpha > 1$. Assuming $m\gamma > \sum_j^d \mu_j^2/\sigma_j^2$, we can set $\alpha = \frac{1}{2} + \frac{m\gamma}{2\sum_j^d \mu_j^2/\sigma_j^2} > 1$ to obtain the bound

$$\mathbb{P}[Z > Z_m] \leq \exp\left(\frac{-\left(m\gamma - \sum_j^d \mu_j^2/\sigma_j^2\right)^2}{8\sum_j^d \mu_j^2/\sigma_j^2}\right). \tag{34}$$

The value of $m$ should be chosen such that this error term is tolerable. For example, if the intent is to obtain an approximate $(\varepsilon, \delta)$-differential privacy bound with $\delta = 10^{-6}$, then we should choose $m$ large enough such that Equation 34 is less than, say, $10^{-9}$.

We should set $m = \frac{1}{\gamma} \cdot \left(\sqrt{8\log(1/\delta')\sum_j^d \mu_j^2/\sigma_j^2} + \sum_j^d \mu_j^2/\sigma_j^2\right)$, where $\delta' > 0$ is the error tolerance in our final estimate of $\delta$.

To obtain lower bounds on $\delta$, we would use

$$\delta = \mathbb{E}\left[\max\{0, 1 - e^{\varepsilon - Z}\}\right] \geq \mathbb{E}\left[\max\{0, 1 - e^{\varepsilon - Z_m}\}\right] - \mathbb{P}[Z < Z_m] \tag{35}$$

and, for all $\alpha > 1$, we have

$$\mathbb{P}[Z < Z_m] = \mathbb{P}[Z \leq -\gamma m/2] \leq \mathbb{E}\left[e^{-\alpha(Z+\gamma m/2)}\right] = e^{(\alpha-1)D_\alpha(M(x')\|M(x))-\alpha\gamma m/2}. \tag{36}$$

11. The algorithm we have sketched above should be relatively efficient and numerically stable. The fast Fourier transform requires $O(m \log m)$ operations. We must evaluate the characteristic function of $Z$ at $m$ points; each evaluation requires evaluating the characteristic function of $d$ discrete Gaussians and multiplying the results together. (Of course, we must only evaluate coordinates where $\mu_j \neq 0$.) The characteristic function of the discrete Gaussian has a very rapidly converging series representation, so this should be close to a constant number of operations.

    The discrete Fourier transform is also numerically stable, since it is a unitary operation. (Indeed this is the advantage of the characteristic function/Fourier transform over the moment generating function/Laplace transform.)

12. The main problem for this algorithm would be if $\gamma$ is extremely small (as the space and time used grows linearly with $1/\gamma$) or if the assumption that $\gamma$ exists fails. This depends on the choice of the parameters $\sigma_1, \cdots, \sigma_d$.

    In this case, one solution is to "bucket" the privacy loss [KJPH20; Goo20a]. That is, rather than relying on the privacy loss naturally falling on a discrete grid $\gamma\mathbb{Z}$ as we do, we artificially round it to such a grid. Rounding up results in computing an upper bound on $\delta$, while rounding down gives a lower bound. The advantage of this bucketing approach is that we have direct control over the granularity of the approximation. The disadvantage is that we cannot use the Poisson summation formula (28) to speed up evaluation of the characteristic function.

## 3   Utility

We now consider how much noise the discrete Gaussian adds. As a comparison point, we consider both the continuous Gaussian and, in the interest of a fair comparison, the rounded Gaussian – i.e., a sample from the continuous Gaussian rounded to the nearest integral value. In Figure 2, we show how these compare numerically. We see that the tail of the rounded Gaussian stochastically dominates that of the discrete Gaussian. In other words, the utility of the discrete Gaussian is strictly better than the rounded Gaussian (although not by much for reasonable values of $\sigma$, i.e., those which are not very small).

To obtain analytic bounds, we begin by bounding the moment generating function:

**Lemma 16.** *Let $t, \sigma \in \mathbb{R}$ with $\sigma > 0$. Then* $\mathop{\mathbb{E}}\limits_{X \leftarrow \mathcal{N}_{\mathbb{Z}}(0,\sigma^2)} \left[ e^{tX} \right] \leq e^{t^2\sigma^2/2}$.

For comparison, recall that the continuous Gaussian satisfies the same bound, but with equality: $\mathop{\mathbb{E}}\limits_{X \leftarrow \mathcal{N}(0,\sigma^2)} \left[ e^{tX} \right] = e^{t^2\sigma^2/2}$ for all $t, \sigma \in \mathbb{R}$ with $\sigma > 0$.

*Proof.* By Lemma 6,

$$
\mathop{\mathbb{E}}\limits_{X \leftarrow \mathcal{N}_{\mathbb{Z}}(0,\sigma^2)} \left[ e^{tX} \right] = \frac{\sum_{x \in \mathbb{Z}} e^{tx - x^2/2\sigma^2}}{\sum_{y \in \mathbb{Z}} e^{-y^2/2\sigma^2}} = \frac{\sum_{x \in \mathbb{Z}} e^{-(x-t\sigma^2)^2/2\sigma^2} \cdot e^{t^2\sigma^2/2}}{\sum_{y \in \mathbb{Z}} e^{-y^2/2\sigma^2}} \leq e^{t^2\sigma^2/2}.
$$

$\square$

Figure 2: Comparison of tail bounds and variance for continuous, discrete, and rounded Gaussians.

The bound on the moment generating function shows that the discrete Gaussian is subgaussian [Riv12]. Standard facts about subgaussian random variables yield bounds on the variance and tails:

**Corollary 17.** *Let $X \leftarrow \mathcal{N}_{\mathbb{Z}}\left(0, \sigma^2\right)$. Then $\mathsf{Var}\left[X\right] \leq \sigma^2$ and $\mathbb{P}\left[X \geq \lambda\right] \leq e^{-\lambda^2/2\sigma^2}$ for all $\lambda \geq 0$.*

Thus the variance of the discrete Gaussian is at most that of the corresponding continuous Gaussian and we also have subgaussian tail bounds. In fact, it is possible to obtain slightly tighter bounds, showing that the variance of the discrete Gaussian is *strictly less* than that of the continuous Gaussian. We elaborate in the following subsections, providing tighter variance and tail bounds. However, these improvements are most pronounced for small $\sigma$, which is not the typical regime of interest for differential privacy. Nonetheless, these facts may be of independent interest.

## 3.1 A Few Good Facts

Here, we state and derive some basic and useful facts about the discrete Gaussian, which will be useful in proving tighter bounds. We start with the expectation of $\mathcal{N}_{\mathbb{Z}}(\mu, \sigma^2)$. It is straightforward to see by a change of index that, for $\mu \in \mathbb{Z}$, one has $\mathbb{E}\left[\mathcal{N}_{\mathbb{Z}}(\mu, \sigma^2)\right] = \mu$; however, the case $\mu \notin \mathbb{Z}$ is not as immediate. Our first result states that, indeed, $\mathcal{N}_{\mathbb{Z}}(\mu, \sigma^2)$ has mean $\mu$ even for non-integer $\mu$:

**Fact 18** (Expectation). *For all $\sigma \in \mathbb{R}$ with $\sigma > 0$, and all $\mu \in \mathbb{R}$, $\mathbb{E}\left[\mathcal{N}_{\mathbb{Z}}(\mu, \sigma^2)\right] = \mu$.*

*Proof.* By the Poisson summation formula [Poi; Wei],

$$\mathbb{E}\left[\mathcal{N}_{\mathbb{Z}}(\mu, \sigma^2)\right] = \frac{\sum_{n\in\mathbb{Z}} n e^{-(n-\mu)^2/(2\sigma^2)}}{\sum_{n\in\mathbb{Z}} e^{-(n-\mu)^2/(2\sigma^2)}} = \frac{\sum_{n\in\mathbb{Z}} \hat{f}(n)}{\sum_{n\in\mathbb{Z}} \hat{g}(n)},$$

where $f(x) = xe^{-(n-\mu)^2/(2\sigma^2)}$ and $g(x) = xe^{-(n-\mu)^2/(2\sigma^2)}$. For $t \in \mathbb{R}$, we can compute their Fourier transforms as

$$\hat{f}(t) = \int_{\mathbb{R}} g(x)e^{-2\pi ixt}\,\mathrm{d}x = \sqrt{2\pi\sigma^2}(\mu - 2\pi i\sigma^2 t)e^{-2\pi^2 t^2\sigma^2 - 2\pi it\mu}$$

$$\hat{g}(t) = \int_{\mathbb{R}} f(x)e^{-2\pi ixt}\,\mathrm{d}x = \sqrt{2\pi\sigma^2}e^{-2\pi^2 t^2\sigma^2 - 2\pi it\mu}$$

so that

$$\mathbb{E}\left[\mathcal{N}_{\mathbb{Z}}(\mu, \sigma^2)\right] = \frac{\mu\sum_{n\in\mathbb{Z}} e^{-2\pi^2 n^2\sigma^2 - 2\pi in\mu} - 2\pi i\sigma^2\sum_{n\in\mathbb{Z}} ne^{-2\pi^2 n^2\sigma^2 - 2\pi in\mu}}{\sum_{n\in\mathbb{Z}} e^{-2\pi^2 n^2\sigma^2 - 2\pi in\mu}} = \mu,$$

as the second sum in the numerator is zero. $\qquad\square$

We now turn to the normalization constant of $\mathcal{N}_{\mathbb{Z}}(0, \sigma^2)$, comparing it to the normalization constant $\sqrt{2\pi\sigma^2}$ of the corresponding continuous Gaussian.

**Fact 19** (Normalization constant). *For all $\sigma \in \mathbb{R}$ with $\sigma > 0$,*

$$\max\{\sqrt{2\pi\sigma^2}, 1\} \leq \sum_{n\in\mathbb{Z}} e^{-n^2/(2\sigma^2)} \leq \sqrt{2\pi\sigma^2} + 1. \tag{37}$$

*Proof.* We first show the lower bound. Clearly $\sum_{n\in\mathbb{Z}} e^{-n^2/(2\sigma^2)} \geq e^{-0^2/2\sigma^2} = 1$. By the Poisson summation formula,

$$\sum_{n\in\mathbb{Z}} e^{-n^2/(2\sigma^2)} = \sum_{n\in\mathbb{Z}} \sqrt{2\pi\sigma^2} \cdot e^{-2\pi^2\sigma^2 n^2} \geq \sqrt{2\pi\sigma^2} \cdot 1.$$

As for the upper bound, it follows from a standard comparison between series and integral:

$$\sum_{n\in\mathbb{Z}} e^{-n^2/(2\sigma^2)} = 1 + 2\sum_{n=1}^{\infty} e^{-n^2/(2\sigma^2)} \leq 1 + 2\sum_{n=1}^{\infty} \int_{n-1}^{n} e^{-x^2/(2\sigma^2)}\mathrm{d}x = 1 + 2\int_{0}^{\infty} e^{-x^2/(2\sigma^2)}\mathrm{d}x.$$

$\qquad\square$

The above bounds, albeit simple to obtain, are not quite as tight as they could be. We state below a refinement, which can be found, e.g., in [Ste17, Claim 2.8.1]:

**Fact 20** (Normalization constant, refined). *For all $\sigma \in \mathbb{R}$ with $\sigma > 0$,*

$$\sqrt{2\pi\sigma^2} \cdot (1 + 2e^{-2\pi^2\sigma^2}) \leq \sum_{n\in\mathbb{Z}} e^{-n^2/(2\sigma^2)} \leq \sqrt{2\pi\sigma^2} \cdot (1 + 2e^{-2\pi^2\sigma^2}) + e^{-2\pi^2\sigma^2} \tag{38}$$

*and*

$$1 + 2e^{-1/(2\sigma^2)} \leq \sum_{n\in\mathbb{Z}} e^{-n^2/(2\sigma^2)} \leq 1 + 2e^{-1/(2\sigma^2)} + \sqrt{2\pi\sigma^2}e^{-1/(2\sigma^2)} \tag{39}$$

*The first set of bounds is better for $\sigma \geq \frac{1}{\sqrt{2\pi}}$, and the second for $\sigma < \frac{1}{\sqrt{2\pi}}$.*

The bounds obtained in Fact 20 are depicted in the figure below.

Figure 3: Bounds from Fact 20 on the normalization constant $\sum_{n\in\mathbb{Z}} e^{-n^2/(2\sigma^2)}$, as a function of $\sigma$. Note that the normalization constant of the continuous Gaussian, $\sqrt{2\pi\sigma^2}$ (in orange) becomes a very accurate approximation for $\sigma \gg 1$; however, for $\sigma \ll 1$, it is not, as the upper and lower bound from Fact 20 both converge towards 1, as expected. Interestingly, we see that the lower bound (green) empirically seems to be nearly tight, as it appears to coincide with the exact expression of the normalization constant (dotted blue) for all $\sigma > 0$. The discontinuity in the upper bound (orange) happens at $\sigma = \frac{1}{\sqrt{2\pi}}$.

## 3.2 Tighter Variance and Tail Bounds

We now analyze the variance of the discrete Gaussian, showing that it is stricty smaller than that of the corresponding continuous Gaussian (and asymptotically the same), with a much better variance for small $\sigma$.

**Proposition 21** (Variance). *For all $\sigma \in \mathbb{R}$ with $\sigma > 0$,*

$$\mathsf{Var}\left[\mathcal{N}_{\mathbb{Z}}(0, \sigma^2)\right] \leq \sigma^2 \left(1 - \frac{4\pi^2\sigma^2}{e^{4\pi^2\sigma^2} - 1}\right) < \sigma^2 \tag{40}$$

*Moreover, if $\sigma^2 \leq 1/3$ then $\mathsf{Var}\left[\mathcal{N}_{\mathbb{Z}}(0, \sigma^2)\right] \leq 3 \cdot e^{-1/2\sigma^2}$.*

To prove Proposition 21 we use the following lemma which relates *upper* bounds on the variance of a discrete Gaussian to *lower* bounds on it, and vice-versa.

**Lemma 22.** *For $\sigma > 0$,*

$$\mathsf{Var}\left[\mathcal{N}_{\mathbb{Z}}(0, \sigma^2)\right] = \sigma^2(1 - 4\pi^2\sigma^2\mathsf{Var}\left[\mathcal{N}_{\mathbb{Z}}(0, 1/(4\pi^2\sigma^2))\right]). \tag{41}$$

*Proof.* By applying the Poisson summation formula to both numerator and denominator of the

variance, we have

$$\mathsf{Var}\left[\mathcal{N}_{\mathbb{Z}}(0,\sigma^2)\right] = \frac{\sum_{n\in\mathbb{Z}} n^2 e^{-n^2/(2\sigma^2)}}{\sum_{n\in\mathbb{Z}} e^{-n^2/(2\sigma^2)}} = \frac{\sum_{n\in\mathbb{Z}} f(n)}{\sum_{n\in\mathbb{Z}} g(n)} = \frac{\sum_{n\in\mathbb{Z}} \hat{f}(n)}{\sum_{n\in\mathbb{Z}} \hat{g}(n)}$$

where $f(x) = x^2 e^{-x^2/(2\sigma^2)}$, $g(x) = e^{-x^2/(2\sigma^2)}$. Now, for $t \in \mathbb{R}$, we can compute

$$\hat{f}(t) = \int_{\mathbb{R}} f(x)e^{-2\pi i x t}\,\mathrm{d}x = \sqrt{2\pi}\sigma^3 e^{-2\pi^2 t^2 \sigma^2}(1 - 4\pi^2 t^2 \sigma^2)$$

$$\hat{g}(t) = \int_{\mathbb{R}} g(x)e^{-2\pi i x t}\,\mathrm{d}x = \sqrt{2\pi}\sigma e^{-2\pi^2 t^2 \sigma^2}.$$

Thus

$$\mathsf{Var}\left[\mathcal{N}_{\mathbb{Z}}(0,\sigma^2)\right] = \sigma^2 \frac{\sum_{n\in\mathbb{Z}} e^{-2\pi^2 n^2 \sigma^2}(1 - 4\pi^2 n^2 \sigma^2)}{\sum_{n\in\mathbb{Z}} e^{-2\pi^2 n^2 \sigma^2}} = \sigma^2 \left(1 - 4\pi^2\sigma^2 \frac{\sum_{n\in\mathbb{Z}} n^2 e^{-2\pi^2 n^2 \sigma^2}}{\sum_{n\in\mathbb{Z}} e^{-2\pi^2 n^2 \sigma^2}}\right)$$

$$= \sigma^2 \left(1 - 4\pi^2\sigma^2 \frac{\sum_{n\in\mathbb{Z}} n^2 e^{-n^2/(2\tau^2)}}{\sum_{n\in\mathbb{Z}} e^{-n^2/(2\tau^2)}}\right) = \sigma^2 \left(1 - 4\pi^2\sigma^2 \mathsf{Var}\left[\mathcal{N}_{\mathbb{Z}}(0,\tau^2)\right]\right)$$

where we set $\tau := \frac{1}{2\pi\sigma}$. $\qquad\square$

Next we have a lower bound on the variance.

**Proposition 23** (Universal Variance Lower Bound). *Let $X$ be a distribution on $\mathbb{R}$ such that* $\mathrm{D}_2\left(X+1\|X\right) \leq \varepsilon^2$. *Then*

$$\mathsf{Var}\left[X\right] \geq \frac{1}{e^{\varepsilon^2} - 1} \tag{42}$$

*Proof.* We follow the proof Lemma C.2 of Bun and Steinke [BS16]. For notational simplicity we assume $X$ has a probability density with respect to the Lesbesgue measure on the reals, which we abusively denote by $\mathbb{P}\left[X = x\right]$. Let $f(x) = \log(\mathbb{P}\left[X+1 = x\right]/\mathbb{P}\left[X = x\right])$ – i.e., $f$ is the logarithm of the Radon-Nikodym derivative of the shifted distribution $X + 1$ with respect to the distribution of $X$. Then

$$e^{\mathrm{D}_2(X+1\|X)} = \int_{\mathbb{R}} \mathbb{P}\left[X+1 = x\right]^2 \mathbb{P}\left[X = x\right]^{-1}\mathrm{d}x = \int_{\mathbb{R}} \left(\frac{\mathbb{P}\left[X+1 = x\right]}{\mathbb{P}\left[X = x\right]}\right)^2 \mathbb{P}\left[X = x\right]\mathrm{d}x = \mathbb{E}\left[e^{2f(X)}\right]$$

and

$$\mathbb{E}\left[X+1\right] = \int_{\mathbb{R}} x\mathbb{P}\left[X+1 = x\right]\mathrm{d}x = \int_{\mathbb{R}} xe^{f(x)}\mathbb{P}\left[X = x\right]\mathrm{d}x = \mathbb{E}\left[X \cdot e^{f(X)}\right].$$

We also have

$$\mathbb{E}\left[e^{f(X)}\right] = \int_{\mathbb{R}} \frac{\mathbb{P}\left[X+1 = x\right]}{\mathbb{P}\left[X = x\right]}\mathbb{P}\left[X = x\right]\mathrm{d}x = \int_{\mathbb{R}} \mathbb{P}\left[X = x\right]\mathrm{d}x = 1.$$

By Cauchy-Schwarz,

$$1 = \mathbb{E}\left[X+1\right] - \mathbb{E}\left[X\right] = \mathbb{E}\left[X \cdot (e^{f(X)} - 1)\right] \leq \sqrt{\mathbb{E}\left[X^2\right] \cdot \mathbb{E}\left[(e^{f(X)} - 1)^2\right]}.$$

This rearranges to give

$$\mathbb{E}\left[X^2\right] \geq \frac{1}{\mathbb{E}\left[(e^{f(X)}-1)^2\right]} = \frac{1}{\mathbb{E}\left[e^{2f(X)}-2e^{f(X)}+1\right]} = \frac{1}{e^{D_2(X+1\|X)}-1}.$$

$\square$

The discrete Gaussian $\mathcal{N}_\mathbb{Z}\left(0, 1/\varepsilon^2\right)$ satisfies the hypotheses of Proposition 23 (by Proposition 5), which yields the following corollary.

**Corollary 24.** *For all $\sigma > 0$,*

$$\underset{X \leftarrow \mathcal{N}_\mathbb{Z}(0,\sigma^2)}{\mathsf{Var}}[X] \geq \frac{1}{e^{1/\sigma^2}-1} \tag{43}$$

We emphasize that the lower bound of Proposition 23 is not specific to the discrete Gaussian. It applies to any distribution $X$ such that adding $X$ to a sensitivity-1 function provides $\frac{1}{2}\varepsilon^2$-concentrated differential privacy.

*Proof of Proposition 21.* Combining Lemma 22 with Proposition 23 (specifically, Corollary 24) yields the first claim:

$$\mathsf{Var}\left[\mathcal{N}_\mathbb{Z}(0,\sigma^2)\right] = \sigma^2(1 - 4\pi^2\sigma^2\mathsf{Var}\left[\mathcal{N}_\mathbb{Z}(0, 1/(4\pi^2\sigma^2))\right]) \leq \sigma^2\left(1 - \frac{4\pi^2\sigma^2}{e^{4\pi^2\sigma^2}-1}\right).$$

Now we establish the last part of the proposition. We have $(m+1)^2 \geq 2m+1$ and, hence,

$$\mathsf{Var}\left[\mathcal{N}_\mathbb{Z}(0,\sigma^2)\right] = \frac{\sum_{n\in\mathbb{Z}} n^2 \cdot e^{-n^2/2\sigma^2}}{\sum_{n\in\mathbb{Z}} e^{-n^2/2\sigma^2}} \leq \sum_{n\in\mathbb{Z}} n^2 \cdot e^{-n^2/2\sigma^2} = 2\sum_{m=0}^{\infty}(m+1)^2 \cdot e^{-(m+1)^2/2\sigma^2}$$

$$\leq 2\sum_{m=0}^{\infty}(m+1)^2 \cdot e^{-(2m+1)/2\sigma^2} = \frac{2}{e^{1/2\sigma^2}}\sum_{m=0}^{\infty}(m+1)^2 e^{-m/\sigma^2}.$$

It only remains to show that $\sum_{m=0}^{\infty}(m+1)^2 e^{-m/\sigma^2} \leq 3/2$ when $\sigma^2 \leq 1/3$. For $x \in (-1,1)$, one can show that $\sum_{m=0}^{\infty}(m+1)^2 x^m = \frac{1+x}{(1-x)^3}$. Set $x = e^{-1/\sigma^2}$ to conclude. $\square$

Next we prove tail bounds:

**Proposition 25.** *For all $m \in \mathbb{Z}$ with $m \geq 1$ and all $\sigma \in \mathbb{R}$ with $\sigma > 0$,*

$$\underset{X \leftarrow \mathcal{N}_\mathbb{Z}(0,\sigma^2)}{\mathbb{P}}[X \geq m] \leq \underset{X \leftarrow \mathcal{N}(0,\sigma^2)}{\mathbb{P}}[X \geq m-1]. \tag{44}$$

*Moreover, if $\sigma \geq 1/\sqrt{2\pi}$, we have*

$$\underset{X \leftarrow \mathcal{N}_\mathbb{Z}(0,\sigma^2)}{\mathbb{P}}[X \geq m] \geq \frac{1}{1 + 3e^{-2\pi^2\sigma^2}} \underset{X \leftarrow \mathcal{N}(0,\sigma^2)}{\mathbb{P}}[X \geq m]. \tag{45}$$

*Proof.* We have $\underset{X \leftarrow \mathcal{N}_\mathbb{Z}(0,\sigma^2)}{\mathbb{P}}[X \geq m] = \frac{\sum_{k=m}^{\infty} e^{-k^2/2\sigma^2}}{\sum_{\ell\in\mathbb{Z}} e^{-\ell^2/2\sigma^2}}$. By Fact 19, the denominator is at least $\sqrt{2\pi\sigma^2}$. For the numerator, we have

$$\sum_{k=m}^{\infty} e^{-k^2/2\sigma^2} = \int_{m-1}^{\infty} e^{-\lceil x\rceil^2/2\sigma^2}\mathrm{d}x \leq \int_{m-1}^{\infty} e^{-x^2/2\sigma^2}\mathrm{d}x = \sqrt{2\pi\sigma^2} \cdot \underset{X \leftarrow \mathcal{N}(0,\sigma^2)}{\mathbb{P}}[X \geq m-1].$$

Turning to the lower bound, suppose $\sigma \geq 1/\sqrt{2\pi}$; by Fact 20, we have $\sum_{\ell \in \mathbb{Z}} e^{-\ell^2/2\sigma^2} \leq \sqrt{2\pi\sigma^2} \cdot (1 + 3e^{-2\pi^2\sigma^2})$, which combined with

$$\sum_{k=m}^{\infty} e^{-k^2/2\sigma^2} = \int_m^{\infty} e^{-\lfloor x \rfloor^2/2\sigma^2} \mathrm{d}x \geq \int_m^{\infty} e^{-x^2/2\sigma^2} \mathrm{d}x = \sqrt{2\pi\sigma^2} \cdot \mathbb{P}_{X \leftarrow \mathcal{N}(0,\sigma^2)}[X \geq m]$$

gives the claim. $\square$

Note that the above proposition focuses on upper tail bounds, but by symmetry of the discrete Gaussian one immediately gets similar lower tail bounds. The upshot is that, up to a small shift or $(1 + o(1))$ multiplicative factor, discrete and continuous Gaussians display the same tails.

One can actually slightly refine the above upper bound, by comparing the discrete Gaussian to the *rounded* Gaussian $\mathcal{N}_{\mathrm{round}}(0, \sigma^2)$, obtained by rounding a standard continuous Gaussian to the nearest integer:

**Proposition 26.** *For all $m \in \mathbb{Z}$ with $m \geq 1$ and all $\sigma \in \mathbb{R}$ with $\sigma > 0$,*

$$\mathbb{P}_{X \leftarrow \mathcal{N}_{\mathbb{Z}}(0,\sigma^2)}[X \geq m] \leq \mathbb{P}_{X \leftarrow \mathcal{N}_{round}(0,\sigma^2)}[X \geq m] + \frac{1}{2} \mathbb{P}_{X \leftarrow \mathcal{N}_{\mathbb{Z}}(0,\sigma^2)}[X = m].$$

*Proof.* On the one hand, by definition of a rounded Gaussian, we have

$$\sqrt{2\pi\sigma^2} \mathbb{P}_{X \leftarrow \mathcal{N}_{\mathrm{round}}(0,\sigma^2)}[X \geq m] = \int_{m-1/2}^{\infty} e^{-x^2/(2\sigma^2)} \mathrm{d}x = \int_{m-1/2}^{m} e^{-x^2/(2\sigma^2)} \mathrm{d}x + \int_m^{\infty} e^{-x^2/(2\sigma^2)} \mathrm{d}x ;$$

on the other hand, we have

$$\sqrt{2\pi\sigma^2} \mathbb{P}_{X \leftarrow \mathcal{N}_{\mathbb{Z}}(0,\sigma^2)}[X \geq m+1] = \sqrt{2\pi\sigma^2} \frac{\sum_{n=m+1}^{\infty} e^{-n^2/(2\sigma^2)}}{\sum_{n \in \mathbb{Z}} e^{-n^2/(2\sigma^2)}} \leq \sum_{n=m+1}^{\infty} e^{-n^2/(2\sigma^2)}$$

by Fact 19. Similarly as before, we can write

$$\sum_{n=m+1}^{\infty} e^{-n^2/(2\sigma^2)} \leq \sum_{n=m+1}^{\infty} \int_{n-1}^{n} e^{-x^2/(2\sigma^2)} \mathrm{d}x = \int_m^{\infty} e^{-x^2/(2\sigma^2)} \mathrm{d}x$$

using monotonicity of $x \mapsto e^{-x^2/(2\sigma^2)}$ on $[0, \infty)$. Combining the three equations above gives

$$\mathbb{P}_{X \leftarrow \mathcal{N}_{\mathbb{Z}}(0,\sigma^2)}[X \geq m] \leq \mathbb{P}_{X \leftarrow \mathcal{N}_{\mathbb{Z}}(0,\sigma^2)}[X = m] + \mathbb{P}_{X \leftarrow \mathcal{N}_{\mathrm{round}}(0,\sigma^2)}[X \geq m] - \frac{1}{\sqrt{2\pi\sigma^2}} \int_{m-1/2}^{m} e^{-x^2/(2\sigma^2)} \mathrm{d}x$$

$$\leq \mathbb{P}_{X \leftarrow \mathcal{N}_{\mathbb{Z}}(0,\sigma^2)}[X = m] + \mathbb{P}_{X \leftarrow \mathcal{N}_{\mathrm{round}}(0,\sigma^2)}[X \geq m] - \frac{1}{2} \frac{e^{-m^2/(2\sigma^2)}}{\sum_{n \in \mathbb{Z}} e^{-n^2/(2\sigma^2)}}$$

$$= \frac{1}{2} \mathbb{P}_{X \leftarrow \mathcal{N}_{\mathbb{Z}}(0,\sigma^2)}[X = m] + \mathbb{P}_{X \leftarrow \mathcal{N}_{\mathrm{round}}(0,\sigma^2)}[X \geq m],$$

as $\int_{m-1/2}^{m} e^{-x^2/(2\sigma^2)} \geq \frac{1}{2} e^{-m^2/(2\sigma^2)}$ and $\sum_{n \in \mathbb{Z}} e^{-n^2/(2\sigma^2)} \geq \sqrt{2\pi\sigma^2}$ by Fact 19. $\square$

We highlight the fact that comparing with the rounded Gaussian, as the above proposition does, is meaningful, since by postprocessing any differential privacy guarantee implied by adding rounded Gaussian noise to discrete data is at least as good as that implied by adding continuous Gaussian noise to the same discrete data.

## 3.3 Other Discretizations, and Convergence to the Continuous Gaussian

Although we focused on this paper on the discrete Gaussian over $\mathbb{Z}$, one can of course consider different discretizations, such as the discrete Gaussian over $\alpha\mathbb{Z} := \{\alpha z : z \in \mathbb{Z}\}$ for some fixed $\alpha > 0$. We denote this distribution by $\mathcal{N}_{\alpha\mathbb{Z}}(\mu, \sigma^2)$. It is defined by

$$\forall x \in \alpha\mathbb{Z} \qquad \underset{X \leftarrow \mathcal{N}_{\alpha\mathbb{Z}}(\mu,\sigma^2)}{\mathbb{P}}[X = x] = \frac{e^{-(x-\mu)^2/2\sigma^2}}{\sum_{y \in \alpha\mathbb{Z}} e^{-(y-\mu)^2/2\sigma^2}}. \tag{46}$$

It is immediate that

$$\forall x \in \alpha\mathbb{Z} \qquad \underset{X \leftarrow \mathcal{N}_{\alpha\mathbb{Z}}(\mu,\sigma^2)}{\mathbb{P}}[X = x] = \underset{X \leftarrow \mathcal{N}_{\mathbb{Z}}(\frac{\mu}{\alpha},\frac{\sigma^2}{\alpha^2})}{\mathbb{P}}\left[X = \frac{x}{\alpha}\right]. \tag{47}$$

In particular, all our results on the (standard) discrete Gaussian will translate to the discrete Gaussian over $\alpha\mathbb{Z}$, up to that change of the parameters $\mu$ and $\sigma$.

Further, one would expect than, as $\alpha \to 0^+$, the discrete Gaussian $\mathcal{N}_{\alpha\mathbb{Z}}(0, \sigma^2)$ converges to $\mathcal{N}(0, \sigma^2)$. We show that this is indeed the case:

**Proposition 27.** *For all $\sigma \in \mathbb{R}$ with $\sigma > 0$, as $\alpha \to 0^+$ the discrete Gaussian $\mathcal{N}_{\alpha\mathbb{Z}}(0, \sigma^2)$ converges in distribution to the continuous Gaussian $\mathcal{N}(0, \sigma^2)$.*

*Proof.* Fix any $0 < \alpha \leq \sqrt{2\pi\sigma^2}$. By Equation 47, for any $x \in \alpha\mathbb{Z}$, we have $\underset{X \leftarrow \mathcal{N}_{\alpha\mathbb{Z}}(0,\sigma^2)}{\mathbb{P}}[X \leq x] = \underset{X \leftarrow \mathcal{N}_{\mathbb{Z}}(0,\frac{\sigma^2}{\alpha^2})}{\mathbb{P}}[X \leq x/\alpha]$, and so, by Proposition 25,

$$\underset{X \leftarrow \mathcal{N}(0,\frac{\sigma^2}{\alpha^2})}{\mathbb{P}}[\alpha X \leq x - \alpha] \leq \underset{X \leftarrow \mathcal{N}_{\alpha\mathbb{Z}}(0,\sigma^2)}{\mathbb{P}}[X \leq x] \leq \frac{1}{1 + 3e^{-2\pi^2\sigma^2/\alpha^2}} \underset{X \leftarrow \mathcal{N}(0,\frac{\sigma^2}{\alpha^2})}{\mathbb{P}}[\alpha X \leq x]$$

or, equivalently,

$$\underset{Y \leftarrow \mathcal{N}(0,\sigma^2)}{\mathbb{P}}[Y \leq x - \alpha] \leq \underset{X \leftarrow \mathcal{N}_{\alpha\mathbb{Z}}(0,\sigma^2)}{\mathbb{P}}[X \leq x] \leq \frac{1}{1 + 3e^{-2\pi^2\sigma^2/\alpha^2}} \underset{Y \leftarrow \mathcal{N}(0,\sigma^2)}{\mathbb{P}}[Y \leq x].$$

Both sides converge to $\underset{Y \leftarrow \mathcal{N}(0,\sigma^2)}{\mathbb{P}}[Y \leq x]$ as $\alpha \to 0^+$. □

In applications where query values are not naturally discrete, it is necessary to round them before adding discrete noise. A finer discretization (i.e., smaller $\alpha$) entails less error being introduced by the rounding.

## 4 Discrete Laplace

We now compare the discrete Gaussian with the most obvious alternative – the discrete Laplace. But first we give a formal definition and state some relevant facts.

**Definition 28** (Discrete Laplace). *Let $t > 0$. The discrete Laplace distribution with scale parameter $t$ is denoted $\mathrm{Lap}_{\mathbb{Z}}(t)$. It is a probability distribution supported on the integers and defined by*

$$\forall x \in \mathbb{Z}, \qquad \underset{X \leftarrow \mathrm{Lap}_{\mathbb{Z}}(t)}{\mathbb{P}}[X = x] = \frac{e^{1/t} - 1}{e^{1/t} + 1} \cdot e^{-|x|/t}. \tag{48}$$

The discrete Laplace (also known as the *two-sided geometric*) was introduced into the differential privacy literature by Ghosh, Roughgarden, and Sundararajan [GRS12], who showed that it satisfies strong optimality properties.

**Lemma 29** (Discrete Laplace Privacy). *Let $\Delta, \varepsilon > 0$. Let $q\colon \mathcal{X}^n \to \mathbb{Z}$ satisfy $|q(x) - q(x')| \leq \Delta$ for all $x, x' \in \mathcal{X}^n$ differing on a single entry. Define a randomized algorithm $M\colon \mathcal{X}^n \to \mathbb{Z}$ by $M(x) = q(x) + Y$ where $Y \leftarrow \mathrm{Lap}_{\mathbb{Z}}(\Delta/\varepsilon)$. Then $M$ satisfies $(\varepsilon, 0)$-differential privacy.*

**Lemma 30** (Discrete Laplace Utility). *Let $\varepsilon > 0$ and let $Y \leftarrow \mathrm{Lap}_{\mathbb{Z}}(1/\varepsilon)$. The distribution is symmetric; in particular, $\mathbb{E}[Y] = 0$. We have $\mathbb{E}[|Y|] = \frac{2 \cdot e^{\varepsilon}}{e^{2\varepsilon}-1}$ and $\mathsf{Var}[Y] = \mathbb{E}[Y^2] = \frac{2 \cdot e^{\varepsilon}}{(e^{\varepsilon}-1)^2}$. For all $\lambda < \varepsilon$,*

$$\mathbb{E}\left[e^{\lambda|Y|}\right] = \frac{e^{\varepsilon}-1}{e^{\varepsilon}+1} \cdot \frac{e^{\varepsilon-\lambda}+1}{e^{\varepsilon-\lambda}-1}.$$

*For all $m \in \mathbb{N}$,*

$$\mathbb{P}[Y \geq m] = \mathbb{P}[Y \leq -m] = \frac{e^{-\varepsilon(m-1)}}{e^{\varepsilon}+1}.$$

We remark that the discrete Laplace can also be efficiently sampled. Indeed, it is a key subroutine of our algorithm for sampling a discrete Gaussian; see Section 5.

There are two immediate qualitative differences between the discrete Laplace and the discrete Gaussian.[8] In terms of utility, the discrete Laplace has subexponential tails (i.e., decaying as $e^{-\varepsilon m}$), whereas the discrete Gaussian has subgaussian tails (i.e., decaying as $e^{-m^2/2\sigma^2}$). In terms of privacy, the discrete Gaussian satisfies concentrated differential privacy, whereas the discrete Laplace satisfies pure differential privacy; pure differential privacy is a qualitatively stronger privacy condition than concentrated differential privacy.

Thus neither distribution dominates the other. They offer different privacy-utility tradeoffs. If the tails are important (e.g., for computing confidence intervals), then the discrete Gaussian is to be favoured. If pure differential privacy is important, then the discrete Laplace is to be favoured.

We now consider a quantitative comparison. To quantify utility, we focus on the variance of the distribution. (An alternative would be to consider the width of a confidence interval.) For now, we will quantify privacy by concentrated differential privacy. Pure $(\varepsilon, 0)$-differential privacy implies $\frac{1}{2}\varepsilon^2$-concentrated differential privacy; thus both distributions can be evaluated on this scale.

Consider a small $\varepsilon > 0$ and a counting query. We can attain $\frac{1}{2}\varepsilon^2$-concentrated differential privacy by adding noise from either $\mathcal{N}_{\mathbb{Z}}\left(0, 1/\varepsilon^2\right)$ or $\mathrm{Lap}_{\mathbb{Z}}(1/\varepsilon)$. By Corollary 17, Proposition 23, and Lemma 30, we have

$$\frac{1}{\varepsilon^2} \geq \underset{Y_{\mathsf{G}} \leftarrow \mathcal{N}_{\mathbb{Z}}(0, 1/\varepsilon^2)}{\mathsf{Var}}[Y_{\mathsf{G}}] \geq \frac{1}{e^{\varepsilon^2}-1} = \frac{1-o(1)}{\varepsilon^2} \qquad \text{and} \qquad \underset{Y_{\mathsf{L}} \leftarrow \mathrm{Lap}_{\mathbb{Z}}(1/\varepsilon)}{\mathsf{Var}}[Y_{\mathsf{L}}] = \frac{2 \cdot e^{\varepsilon}}{(e^{\varepsilon}-1)^2} = \frac{2 \pm o(1)}{\varepsilon^2}.$$

Thus, asymptotically (i.e., for small $\varepsilon$), the discrete Gaussian has half as much variance as the discrete Laplace for the same level of privacy. In this comparison, the Gaussian clearly is better.

However, the above quantitative comparison is potentially unfair. Quantifying differential privacy by concentrated differential privacy may favour the Gaussian. If instead we demand pure $(\varepsilon, 0)$-differential privacy or approximate $(\varepsilon, \delta)$-differential privacy for a small $\delta > 0$, then the comparison would yield the opposite conclusion. It is fundamentally difficult to compare algorithms satisfying different versions of differential privacy, as there is no level playing field.

There is another factor to consider: A practical differentially private system will answer many queries via independent noise addition. Thus the real object of interest is the privacy and utility of the *composition* of many applications of noise addition.

For the rest of this section, we consider the task of answering $k$ counting queries (or sensitivity-1 queries) by adding either discrete Gaussian or discrete Laplace noise. We will measure privacy by approximate $(\varepsilon, \delta)$-differential privacy over a range of parameters. The results are summarized in Figure 4.

Concentrated differential privacy has an especially clean composition theorem [BS16]:

**Lemma 31** (Composition for Concentrated Differential Privacy). *Let* $M_1 \colon \mathcal{X}^n \to \mathcal{Y}_1$ *satisfy* $\frac{1}{2}\varepsilon_1^2$-*concentrated differential privacy. Let* $M_2 \colon \mathcal{X}^n \times \mathcal{Y}_1 \to \mathcal{Y}_2$ *be such that, for all* $y \in \mathcal{Y}_1$, *the restriction* $M_2(\cdot, y) \colon \mathcal{X}^n \to \mathcal{Y}_2$ *satisfies* $\frac{1}{2}\varepsilon_2^2$-*concentrated differential privacy. Define* $M_* \colon \mathcal{X}^n \to \mathcal{Y}_2$ *by* $M_*(x) = M_2(x, M_1(x))$. *Then* $M_*$ *satisfies* $\frac{1}{2}(\varepsilon_1^2 + \varepsilon_2^2)$-*concentrated differential privacy.*

This result can be extended to $k$ mechanisms by induction. Thus, to attain $\frac{1}{2}\varepsilon^2$-concentrated differential privacy for $k$ counting queries, it suffices to add noise from $\mathcal{N}_{\mathbb{Z}}\left(0, k/\varepsilon^2\right)$ to each value independently. We then convert the overall $\frac{1}{2}\varepsilon^2$-concentrated differential privacy guarantee into approximate $(\varepsilon', \delta)$-differential privacy using Corollary 13.

In contrast, analysing the composition of multiple invocations of discrete Laplace noise addition is not as clean. We use an optimal composition result provided by Kairouz, Oh, and Viswanath [KOV17; MV16]: The $k$-fold composition of $(\varepsilon, \delta)$-differential privacy satisfies $(\varepsilon', \delta')$-differential privacy if and only if

$$\frac{1}{(1 + e^\varepsilon)^k} \sum_{\ell=0}^{k} \binom{k}{\ell} \max\left\{0, e^{\ell\varepsilon} - e^{\varepsilon' + (k-\ell)\varepsilon}\right\} \leq 1 - \frac{1 - \delta'}{(1 - \delta)^k}. \tag{49}$$

Figure 4: Comparison of discrete Gaussian and Laplace noise addition. Left: Utility is fixed (i.e., answer $k = 100$ counting queries each with variance $50^2$ )and we consider the curve of approximate $(\varepsilon, \delta)$-differential privacy guarantees that we can achieve. Right: Privacy is fixed (i.e., approximate $(1, 10^{-6})$-differential privacy) and we consider the utility (i.e., variance of noise added to each answer) as we vary the number of counting queries to be answered.

In Figure 4, we compare the discrete Gaussian and the discrete Laplace in two ways. First (on the left), we fix the utility and compare the approximate differential privacy guarantees. Specifically, we fix the task of answering $k = 100$ counting queries with the noise added to each value having variance $50^2$. Both distributions yield different curves of $(\varepsilon, \delta)$-differential privacy guarantees and there are many points to consider. We see that, for this task, the discrete Gaussian attains better $(\varepsilon, \delta)$-differential privacy guarantees except for extremely small $\delta$ – specifically, $\delta < 10^{-45}$. For $\varepsilon = 1$, the discrete Gaussian provides $(1, 10^{-7})$-differential privacy for this task, whereas the discrete Laplace only provides $(1, 206 \times 10^{-7})$-differential privacy. If we demand pure differential privacy, then the discrete Laplace provides $(2.83, 0)$-differential privacy, but the discrete Gaussian cannot provide pure differential privacy. The separation becomes more pronounced as the number of queries grows.

Second (on the right of Figure 4), we fix the privacy goal to approximate $(1, 10^{-6})$-differential privacy. We vary the number of counting queries (from $k = 1$ to $k = 100$) and measure the variance of the noise that must be added to each query answer. For a small number of queries ($k \leq 10$), the discrete Laplace gives lower variance. However, as the number of queries increases, we see that the discrete Laplace requires higher variance; for $k = 100$, the variance is 69% more.

Overall, Figure 4 demonstrates that the discrete Gaussian provides a better privacy-utility tradeoff than the discrete Laplace, except in two narrow parameter regimes: Either a small number of queries or if we demand something very close to pure differential privacy. We only compared variances; if we compare confidence interval sizes instead, then this would further advantage the Gaussian, which has lighter tails.

# 5 Sampling

In this section, we show how to efficiently sample exactly from a discrete Gaussian on a finite computer given access only to uniformly random bits. Such algorithms are already known [Kar16; DFW20]. However, we include this both for completeness because we believe that our algorithms are simpler than the prior work. A sample `Python` implementation is available online [Dga].

For simplicity, we focus our discussion of runtime only on the expected number of arithmetic operations; each such operation will take time polylogarithmic in the bit complexity of the parameters (e.g., in the representation of $\sigma^2$ as a rational number). We elaborate on this at the end of the section.

In Algorithm 3, we present a simple and fast algorithm for discrete Gaussian sampling, with the following guarantees:

**Theorem 32.** *On input $\sigma^2 \in \mathbb{Q}$, the procedure described in Algorithm 3 outputs one sample from $\mathcal{N}_{\mathbb{Z}}\left(0, \sigma^2\right)$ and requires only a constant number of operations in expectation.*

At a high level, the idea behind the algorithm is to first sample from a discrete Laplace distribution and then "convert" this into a discrete Gaussian by rejection sampling. In order to do so, we provide two subroutines, which we believe to be of independent interest: the first, to efficiently and exactly sample from a Bernoulli with parameter $e^{-\gamma}$, for any rational parameter $\gamma \geq 0$ (Proposition 33). The second, to efficiently and exactly sample from a discrete Laplace with scale parameter $t$, for any positive integer $t$ (Proposition 34).

## 5.1   Sampling $\mathsf{Bernoulli}(\exp(-\gamma))$

Our first subroutine, Algorithm 1, describes how to reduce the task of sampling from $\mathsf{Bernoulli}(\exp(-\gamma))$ to that of sampling from $\mathsf{Bernoulli}(\gamma/k)$ for various integers $k \geq 1$. This procedure is based on a technique of von Neumann [VN51; For72]. This procedure avoids complex operations, such as computing the exponential function. Thus, for a rational $\gamma$, this can be implemented on a finite computer. Specifically, for $n, d \in \mathbb{N}$, to sample $\mathsf{Bernoulli}(n/d)$ it suffices to draw $D \in \{1, 2, \ldots, d\}$ uniformly at random and output 1 if $D \leq n$ and output 0 if $D > n$. (To sample $D \in \{1, 2, \cdots, d\}$ we can again use rejection sampling – that is, we uniformly sample $D \in \{1, 2, \cdots, 2^{\lceil \log_2 d \rceil}\}$ and reject and retry if $D > d$.)

In the rest of the analysis, we assume for the sake of abstraction that sampling $\mathsf{Bernoulli}(n/d)$ given $n, d \in \mathbb{N}$ requires a constant number of arithmetic operations in expectation.

**Proposition 33.** *On input (rational) $\gamma \geq 0$, the procedure described in Algorithm 1 outputs one sample from $\mathsf{Bernoulli}(\exp(-\gamma))$, and requires a constant number of operations in expectation.*

*Proof.* First, consider the case where $\gamma \in [0, 1]$. For the analysis, we let $A_k$ denote the value of $A$ in the $k$-th iteration of the loop in the algorithm, and $K^*$ denote the final value of $K$ upon exiting the loop. Then, for all $k \in \{0, 1, 2, \cdots\}$, we have

$$\mathbb{P}\left[K^* > k\right] = \mathbb{P}\left[A_1 = A_2 = \cdots = A_k = 1\right] = \prod_{i=1}^{k} \mathbb{P}\left[A_i = 1\right] = \prod_{i=1}^{k} \frac{\gamma}{i} = \frac{\gamma^k}{k!}.$$

Thus

$$\mathbb{P}\left[K^* \text{ odd}\right] = \sum_{k=0}^{\infty} \mathbb{P}\left[K^* = 2k + 1\right] = \sum_{k=0}^{\infty} (\mathbb{P}\left[K^* > 2k\right] - \mathbb{P}\left[K^* > 2k + 1\right]) = \sum_{k=0}^{\infty} \left(\frac{\gamma^{2k}}{(2k)!} - \frac{\gamma^{2k+1}}{(2k+1)!}\right)$$

which is equal to $e^{-\gamma}$ as desired. Further, the expected number of operations is simply $T(\gamma) = O(\mathbb{E}\,[K^*]) = O(\sum_{k=0}^{\infty} \mathbb{P}\,[K^* > k]) = O(e^{\gamma}) = O(1)$.

Now, if $\gamma > 1$, the procedure performs (at most) $\ell := \lfloor \gamma \rfloor + 1$ independent sequential recursive calls, getting $\ell$ independent samples $B_1, \cdots, B_{\ell-1} \sim \mathsf{Bernoulli}(\exp(-1))$ and $C \sim \mathsf{Bernoulli}(\exp(-(\gamma - \lfloor \gamma \rfloor)))$. Its output is then distributed as a Bernoulli random variable with parameter $\mathbb{P}\,[B_1 = B_2 = \cdots = B_{\ell-1} = C = 1] = \prod_{i=1}^{\ell} \mathbb{P}\,[B_i = 1] \cdot \mathbb{P}\,[C = 1] = \exp(-1)^{\lfloor \gamma \rfloor} \cdot \exp(\gamma - \lfloor \gamma \rfloor) = \exp(-\gamma)$, as desired.

The number of recursive calls is at most $\ell$. However, the recursive calls will stop as soon as $B = 0$ for the first time. We have $\mathbb{P}\,[B = 0] = 1 - e^{-1}$. Thus the number of recursive calls follows a truncated geometric distribution. The expected number of recursive calls is constant and, therefore, the expected number of operations is too. $\qquad\square$

---

**Algorithm 1** Algorithm for Sampling $\mathsf{Bernoulli}(\exp(-\gamma))$.

---

**Input:** Parameter $\gamma \geq 0$.
**Output:** One sample from $\mathsf{Bernoulli}(\exp(-\gamma))$.
  **if** $\gamma \in [0, 1]$ **then**
     Set $K \leftarrow 1$.
     **loop**
        Sample $A \leftarrow \mathsf{Bernoulli}(\gamma/K)$.
        **if** $A = 0$ **then** break the loop.
        **if** $A = 1$ **then** set $K \leftarrow K + 1$ and continue the loop.
     **if** $K$ is odd **then return** 1.
     **if** $K$ is even **then return** 0.
  **else**
     **for** $k = 1$ **to** $\lfloor \gamma \rfloor$ **do**
        Sample $B \leftarrow \mathsf{Bernoulli}(\exp(-1))$                  $\triangleright$ Recursive call.
        **if** $B = 0$ **then** break the loop and **return** 0.
     Sample $C \leftarrow \mathsf{Bernoulli}(\exp(\lfloor \gamma \rfloor - \gamma))$    $\triangleright$ Recursive call. $\gamma - \lfloor \gamma \rfloor \in [0, 1]$.
     **return** $C$.

---

## 5.2 Sampling from a Discrete Laplace

Now we show how to efficiently and exactly sample from a discrete Laplace distribution; see Section 4 for more about this distribution. Other methods for sampling from the discrete Laplace distribution are known [SWSZW19].

**Proposition 34.** *On input $s, t \in \mathbb{Z}$ with $s, t \geq 1$, the procedure described in Algorithm 2 outputs one sample from $\mathrm{Lap}_{\mathbb{Z}}(t/s)$, and requires a constant number of operations in expectation.*

*Proof.* To prove the theorem, we must verify two things: (1) we must show that, for each attempt (i.e., for each iteration of the outer loop), conditioned on outputting a value $Z$ (henceforth referred to as *success*, and denoted $\top$), the distribution of the output $Z$ is $\mathrm{Lap}_{\mathbb{Z}}(t/s)$ as desired; (2) we must lower bound the probability that a given loop iteration is successful. This ensures that the loop terminates quickly, giving the bound on the runtime.

---
**Algorithm 2** Algorithm for Sampling a Discrete Laplace
---
**Input:** Parameters $s, t \in \mathbb{Z}$, $s, t \geq 1$.
**Output:** One sample from $\mathrm{Lap}_{\mathbb{Z}}(t/s)$.

   **loop**                                                                                           ▷ Repeat until successful
      Sample $U \in \{0, 1, 2, \cdots, t-1\}$ uniformly at random.
      Sample $D \leftarrow \mathsf{Bernoulli}(\exp(-U/t))$.                                  ▷ Use Algorithm 1.
      **if** $D = 0$ **then** reject and restart.
      Initialize $V \leftarrow 0$.
      **loop**                                          ▷ Generate $V$ from $\mathsf{Geometric}(1 - e^{-1})$.
         Sample $A \leftarrow \mathsf{Bernoulli}(\exp(-1))$.                       ▷ Use Algorithm 1.
         **if** $A = 0$ **then** break the loop.
         **if** $A = 1$ **then** set $V \leftarrow V + 1$ and continue.
      Set $X \leftarrow U + t \cdot V$.                              ▷ $X$ is $\mathsf{Geometric}(1 - e^{-1/t})$.
      Set $Y \leftarrow \lfloor X/s \rfloor$                              ▷ $Y$ is $\mathsf{Geometric}(1 - e^{-s/t})$.
      Sample $B \leftarrow \mathsf{Bernoulli}(1/2)$.
      **if** $B = 1$ and $Y = 0$ **then** reject and restart.
      **return** $Z \leftarrow (1 - 2B) \cdot Y$.                       ▷ Success; $Z$ is a discrete Laplace.
---

To begin, we show that, conditioned on $D = 1$, $X$ follows a geometric distribution with parameter $\tau := 1/t$. Specifically, $\mathbb{P}[X = x \mid D = 1] = (1 - e^{-\tau}) \cdot e^{-x\tau}$ for every integer $x \geq 0$. For any such $x$, let $u_x := x \bmod t$ and $v_x = \lfloor x/t \rfloor$, so that $x = u_x + t \cdot v_x$. It is immediate to see that, as defined by the algorithm, $V$ is independent of both $U$ and $D$ and follows a geometric distribution with parameter $1 - e^{-1}$: that is, $\mathbb{P}[V = k] = (1 - e^{-1}) \cdot e^{-k}$ for every integer $k \geq 0$. We thus have

$$\mathbb{P}[X = x \mid D = 1] = \mathbb{P}[U = u_x, V = v_x \mid D = 1] = \mathbb{P}[U = u_x \mid D = 1] \cdot \mathbb{P}[V = v_x]$$

$$= \frac{\mathbb{P}[U = u_x]}{\mathbb{P}[D = 1]} \cdot \mathbb{P}[D = 1 \mid U = u_x] \cdot (1 - e^{-1}) \cdot e^{-v_x}$$

$$= \frac{1/t}{(1/t) \sum_{k=0}^{t-1} e^{-k/t}} \cdot e^{-u_x/t} \cdot (1 - e^{-1}) \cdot e^{-v_x}$$

$$= (1 - e^{-1/t}) \cdot e^{-(u_x/t + v_x)} = (1 - e^{-1/t}) \cdot e^{-x/t}$$

as claimed.

We then claim that $Y = \lfloor \frac{X}{s} \rfloor$ (conditioned on $D = 1$) follows a $\mathsf{Geometric}(1 - e^{-s/t})$ distribution – i.e., $\mathbb{P}[Y = y \mid D = 1] = (1 - e^{-s/t}) \cdot e^{-y \cdot s/t}$ for all integers $y \geq 0$. This is an immediate consequence of the following fact.

**Fact 35.** *Fix $p \in (0, 1]$. Let $G$ be a $\mathsf{Geometric}(1 - p)$ random variable, and $n \geq 1$ be an integer. Then $\lfloor \frac{G}{n} \rfloor$ is a $\mathsf{Geometric}(1 - q)$ random variable for $q = p^n$.*

*Proof.* For any integer $k \geq 0$,

$$\mathbb{P}[\lfloor G/n \rfloor = k] = \mathbb{P}[nk \leq G < (k+1)n] = \sum_{\ell=kn}^{(k+1)n-1} (1 - p)p^\ell = (1 - p^n)p^{nk} = (1 - q)q^k.$$

$\square$

With this in hand, we analyze the distribution of $Z$ conditioned on $\top$ (success), i.e., conditioned on $D = 1$ and $(B, Y) \neq (1, 0)$.[9] Let $\alpha := s/t$ for convenience. Recalling $B$ is independent of $D$ and that $B$ and $Y$ (conditioned on $D = 1$) are independent, we have

$$\mathbb{P}\left[(B, Y) \neq (1, 0) \mid D = 1\right] = \mathbb{P}\left[B = 1, Y > 0 \mid D = 1\right] + \mathbb{P}\left[B = 0 \mid D = 1\right] = \frac{1}{2}(e^{-\alpha} + 1).$$

For every $z \in \mathbb{Z}$, we have, recalling that $B$ and $Y$ are independent,

$$
\begin{aligned}
\mathbb{P}\left[Z = z \mid \top\right] &= \mathbb{P}\left[(1 - 2B)Y = z \mid (B, Y) \neq (1, 0), D = 1\right] \\
&= \frac{\mathbb{P}\left[(1 - 2B)Y = z, (B, Y) \neq (1, 0) \mid D = 1\right]}{\mathbb{P}\left[(B, Y) \neq (1, 0) \mid D = 1\right]} \\
&= \frac{\mathbb{P}\left[Y = |z|, B = \mathbb{I}[z < 0] \mid D = 1\right]}{\mathbb{P}\left[(B, Y) \neq (1, 0) \mid D = 1\right]} \\
&= \frac{\mathbb{P}\left[Y = |z| \mid D = 1\right] \cdot \mathbb{P}\left[B = \mathbb{I}[z < 0] \mid D = 1\right]}{\mathbb{P}\left[(B, Y) \neq (1, 0) \mid D = 1\right]} \\
&= \frac{\mathbb{P}\left[Y = |z| \mid D = 1\right]}{2\mathbb{P}\left[(B, Y) \neq (1, 0) \mid D = 1\right]} \\
&= \frac{(1 - e^{-\alpha}) \cdot e^{-|z| \cdot \alpha}}{e^{-\alpha} + 1},
\end{aligned}
$$

by our previous computations. Thus, conditioned on success, $Z$ follows a $\mathrm{Lap}_{\mathbb{Z}}(1/\alpha)$ distribution.

We then bound the probability that a fixed iteration of the loop succeeds:

$$\mathbb{P}\left[\top\right] = \mathbb{P}\left[(B, Y) \neq (1, 0) \mid D = 1\right]\mathbb{P}\left[D = 1\right] = \frac{1 + e^{-\alpha}}{2} \cdot \frac{1}{t}\sum_{u=0}^{t-1} e^{-u/t} = \frac{1 + e^{-\alpha}}{2}\frac{1 - e^{-1}}{t(1 - e^{-1/t})} \geq \frac{1 - e^{-1}}{2}.$$

It follows that the number $N$ of iterations of the outer loop needed to output a value $Z$ is geometrically distributed and satisfies $\mathbb{E}\left[N\right] \leq \frac{2}{1 - e^{-1}} < 3.2$. Moreover, each iteration of the outer loop requires a constant number of operations in expectations. This is because each of the subroutines requires a constant number of operations in expectation and the inner loop runs a geometrically-distributed number of times which is constant in expectation. $\qquad\square$

## 5.3 Sampling from a Discrete Gaussian

In Algorithm 3, we prove Theorem 32 and present our algorithm that requires $O(1)$ operations on average to sample from a discrete Gaussian $\mathcal{N}_{\mathbb{Z}}\left(0, \sigma^2\right)$.

---

**Algorithm 3** Algorithm for Sampling a Discrete Gaussian

---

**Input:** Parameter $\sigma^2 > 0$.
**Output:** One sample from $\mathcal{N}_{\mathbb{Z}}\left(0, \sigma^2\right)$.
  Set $t \leftarrow \lfloor \sigma \rfloor + 1$
  **loop**                                                           $\triangleright$ Repeat until successful
    Sample $Y \leftarrow \mathrm{Lap}_{\mathbb{Z}}(t)$                                 $\triangleright$ Use Algorithm 2
    Sample $C \leftarrow \mathsf{Bernoulli}(\exp(-(|Y| - \sigma^2/t)^2/2\sigma^2))$.         $\triangleright$ Use Algorithm 1
    If $C = 0$, reject and restart.
    If $C = 1$, return $Y$ as output.         $\triangleright$ Success; $Y$ is a discrete Gaussian.

*Proof of Theorem 32.* Fix any iteration of the loop, and let $t \leftarrow \lfloor \sigma \rfloor + 1$ and $\tau := 1/t$. Since $Y \leftarrow \mathrm{Lap}_{\mathbb{Z}}(1/\tau)$, we have that $C$ is a Bernoulli with parameter

$$\mathbb{E}\left[C\right] = \mathbb{E}\left[\mathbb{E}\left[C \mid Y\right]\right] = \mathbb{E}\left[e^{-\frac{(|Y|-\sigma^2\tau)^2}{2\sigma^2}}\right] = \frac{1-e^{-\tau}}{1+e^{-\tau}}\sum_{y \in \mathbb{Z}} e^{-\frac{(|y|-\sigma^2\tau)^2}{2\sigma^2}-|y|\tau} = \frac{1-e^{-\tau}}{1+e^{-\tau}}e^{-\frac{\sigma^2\tau^2}{2}}\sum_{y \in \mathbb{Z}}e^{-\frac{y^2}{2\sigma^2}}.$$

Thus, for any $y \in \mathbb{Z}$, conditioned on $C = 1$ (i.e., on $Y$ being output) we have

$$\mathbb{P}\left[Y = y \mid C = 1\right] = \frac{\mathbb{P}\left[C = 1 \mid Y = y\right]\mathbb{P}\left[Y = y\right]}{\mathbb{P}\left[C = 1\right]} = \frac{e^{-\frac{(|y|-\sigma^2\tau)^2}{2\sigma^2}} \cdot \frac{1-e^{-\tau}}{1+e^{-\tau}} \cdot e^{-|y|\tau}}{\mathbb{E}\left[C\right]}$$

$$= \frac{e^{-\frac{(|y|-\sigma^2\tau)^2}{2\sigma^2}} \cdot e^{-|y|\tau}}{e^{-\frac{\sigma^2\tau^2}{2}}\sum_{y' \in \mathbb{Z}}e^{-\frac{y'^2}{2\sigma^2}}} = \frac{e^{-\frac{y^2}{2\sigma^2}}}{\sum_{y' \in \mathbb{Z}}e^{-\frac{y'^2}{2\sigma^2}}}.$$

That is, conditioned on outputting a value, this value is indeed distributed according to $\mathcal{N}_{\mathbb{Z}}\left(0, \sigma^2\right)$.

We now turn to the runtime analysis. First, recalling (1) that $\sigma^2\tau^2 < 1$ and $\sigma \geq t - 1$, by our choice of $t = 1/\tau = \lfloor \sigma \rfloor + 1 > \sigma$, and (2) the bound $\sum_{y \in \mathbb{Z}} e^{-\frac{y^2}{2\sigma^2}} \geq \max\{1, \sqrt{2\pi\sigma^2}\}$ from Fact 19, we have

$$\mathbb{E}\left[C\right] \geq \frac{1-e^{-1/t}}{1+e^{-1/t}}e^{-\frac{1}{2}}\max\{1, \sqrt{2\pi}\sigma\} \geq \frac{e^{-1/2}\sqrt{2\pi}}{2}(1 - e^{-1/t})\max\{1, t - 1\} > 0.29.$$

Therefore the probability that the algorithms succeeds and outputs a value in any given iteration of the loop is lower bounded by a positive constant. Thus the number of iterations of the loop follows a geometric distribution and is constant in expectation. Since, for each iteration, the expected number of operations required to sample $Y$ and $C$ is constant (by Propositions 34 and 33) the overall number of operations is constant in expectation. $\square$

## 5.4 Runtime Analysis

We have stated that our algorithms require a constant number of operations in expectation. We now elaborate on this.

We assume a Word RAM model of computation. In particular, we assume that arithmetic operations on the parameters count as one operation. Specifically, we assume that the parameter $\sigma^2$ is represented as a rational number (i.e., two integers in binary) and that this fits in a constant number of words. If we measure complexity in terms of bits (rather than words), then all operations run in time polynomial in the description length of the input $\sigma^2$. We emphasize that, if the parameter $\sigma^2$ is rational, then all operations are over rational numbers; we only apply basic field operations and comparisons and do not evaluate any functions like the exponential function or the square root[10] that would require approximations or moving outside the rational field. The memory (i.e., number of words) used by our algorithms is logarithmic in the runtime and constant in expectation. (The only way the memory usage grows is the counters associated with some loops.)

The runtime of our algorithms is random. Beyond showing that the number of operations is constant in expectation, it is possible to show, for all of our algorithms, that it is a subexponential random variable. We give a precise definition of this term.

**Definition 36.** *A nonnegative random variable $X$ is said to be $\lambda$-subexponential if $\mathbb{E}\left[e^{X/\lambda}\right] \leq e$. And $X$ is said to be* subexponential *if it is $\lambda$-subexponential for some finite $\lambda > 0$.*

The constant $e$ in the definition is arbitrary. Note that, if $X$ is $\lambda$-subexponential, then $\mathbb{P}\left[X \geq t\right] \leq \mathbb{E}\left[e^{(X-t)/\lambda}\right] \leq e^{1-t/\lambda}$ for all $t \geq 0$.

Our algorithms effectively consist of a constant number of nested loops and the number of times each of them runs is subexponential. For most of our loops, they have a constant probability of terminating in each run, which means the number of times they run follows a geometric distribution, which is a subexponential random variable.

It turns out that such nested loops also have a subexpoential runtime. Specifically, one can show that, if $X_1, \ldots, X_n, \ldots$ are independent subexponential random variables and $T$ is a stopping time that is subexponential, then $\sum_{n=1}^{T} X_n$ is still subexponential:

**Lemma 37.** *Let $\alpha, \beta > 1$. Suppose $(X_n)_{1 \leq n \leq \infty}$ are independent non-negative $\alpha$-subexponential random variables and $T$ is a $\beta$-subexponential stopping time. Then $S := \sum_{n=1}^{T} X_n$ is $\alpha\beta$-subexponential.*

*Proof.* We will require the following simple result:

**Claim 38.** *Let $(Y_n)_{n \geq 1}$ be independent random variables satisfying $\mathbb{E}[e^{Y_n}] \leq 1$ for all $n$, and let $T$ be a stopping time such that $T < \infty$ almost surely. Then $\mathbb{E}[e^{\sum_{n=1}^{T} Y_n}] \leq 1$.*

*Proof.* For $n \geq 0$, let $M_n := e^{\sum_{k=1}^{n} Y_k} \geq 0$ (so that $M_0 = 1$). Note that $\mathbb{E}[M_{n+1} \mid M_1, \ldots, M_n] = \mathbb{E}[e^{X_{n+1}}]M_n \leq M_n$, i.e., $(M_n)_{n \geq 0}$ is a supermartingale. By the optional stopping theorem for non-negative supermartingales (cf., e.g., [Wil91, Corollary 10.10(d)]), as $T$ is a.s. finite we get $\mathbb{E}[M_T] \leq \mathbb{E}[M_0] = 1$, that is, $\mathbb{E}[e^{\sum_{n=1}^{T} Y_n}] \leq 1$. $\qquad\square$

Applying Claim 38 to $Y_n := X_n/\alpha - 1$, we get $\mathbb{E}\left[e^{S/\alpha - T}\right] = \mathbb{E}[e^{\sum_{n=1}^{T} X_n/\alpha - T}] \leq 1$. Now, by Hölder's and Jensen's inequalities,

$$\mathbb{E}\left[e^{S/\alpha\beta}\right] = \mathbb{E}\left[e^{(S/\alpha - T)/\beta} \cdot e^{T/\beta}\right] \leq \mathbb{E}\left[e^{S/\alpha - T}\right]^{1/\beta} \cdot \mathbb{E}\left[e^{T/(\beta-1)}\right]^{1-1/\beta} \leq 1^{1/\beta} \cdot \mathbb{E}\left[e^{T/\beta}\right] \leq e.$$

Thus $S$ is $\alpha\beta$-subexponential. $\qquad\square$

In our case, $T$ corresponds to the number of times the loop runs and $X_n$ corresponds to the number of operations required inside the $n$-th run of the loop. Applying the above lemma to each nested loop shows that the overall runtimes of Algorithm 1, Algorithm 2, and Algorithm 3 are all subexponential random variables.

This means in particular that, for any $\delta \in (0,1)$, to generate $k$ samples from $\mathcal{N}_\mathbb{Z}\left(0, \sigma^2\right)$ (i.e., $k$ runs of Algorithm 3), the probability of requiring more than $O(k + \log(1/\delta))$ operations is at most $\delta$. (If $X_1, \cdots, X_k$ are the number of operations of the $k$ runs, then $\mathbb{P}\left[X_1 + \cdots + X_k \geq t\right] \leq \mathbb{E}\left[e^{(X_1 + \cdots + X_k - t)/\lambda}\right] \leq e^{k - t/\lambda}$, where $\lambda$ is the subexponential constant of Algorithm 3's number of operations. Setting $t = \lambda(k + \log(1/\delta))$ ensures this probability is at most $\delta$.)

Thus, our algorithms have a highly concentrated runtime. This is important: If they do not terminate in time, then this may result in a failure of differential privacy. There is also the potential for timing attacks. Balcer and Vadhan [BV17] argue that differentially private algorithms should have a deterministic running time to avoid these issues altogether. However, this is a highly restrictive model. We cannot exactly sample the discrete Gaussian (or any unbounded distribution) in this model. It is not even possible to exactly sample from Bernoulli$(1/3)$ in this model (since,

if we only have access to $\ell$ random bits, we can only generate probabilities that are a multiple of $2^{-\ell}$).

By terminating (and outputting 0) after a pre-specified time limit, our algorithms can be made to have a deterministic runtime. However, this comes at the expense of now only satisfying approximate $(\varepsilon, \delta + \delta')$-differential privacy or $\delta'$-approximate $\frac{1}{2}\varepsilon^2$-concentrated differential privacy [BS16], where $\delta'$ is the probability of reaching the time limit. Since the running time is roughly subexponential, this failure probability $\delta'$ can be made astronomically small with no cost in accuracy and very little cost in runtime (i.e., only milliseconds overall). Realistically, a far greater concern than this failure probability is that the source of random bits is not perfectly uniform [GL20].

**Practical remark.** We have implemented the algorithms from Algorithms 1, 2, and 3 in `Python` (using the `fractions.Fraction` class for exact rational arithmetic and using `random.SystemRandom()` to obtain high-quality randomnesss). Overall, on a standard personal computer, our basic (non-optimized) implementation is able to produce over 1000 samples per second even for $\sigma^2 = 10^{100}$. The source code is available online [Dga].

# Acknowledgments

We thank Shahab Asoodeh, Damien Desfontaines, Peter Kairouz, and Ananda Theertha Suresh for making us aware of several related works.

## Footnotes

[1] Dinur and Nissim [DN03] also analyzed the privacy properties of Binomial noise addition, but this predates the definition of differential privacy.

[2] We take log to be the natural logarithm – i.e., base $e \approx 2.718$.

[3] We use the parameterization $\frac{1}{2}\varepsilon^2$-concentrated differential privacy instead of $\rho$-concentrated differential privacy as in the original paper. This is because $\varepsilon$ is a more familiar privacy parameter and, by setting $\rho = \frac{1}{2}\varepsilon^2$, we put it on the same "scale" as pure or approximate differential privacy. We revert to $\rho$ where it might otherwise be confusing, e.g., in Corollary 13 where we simultaneously discuss concentrated differential privacy and approximate differential privacy.

[4]In the information theory literature, the term "relative information spectrum" is sometimes used for the distribution of what we call the privacy loss random variable [SV16; Liu18].

[5]Here we abuse notation: We use notation that only is well-defined for discrete random variables. However, the result holds in general under appropriate assumptions.

[6]This assumption is the definition of $(\alpha, \tau)$-Rényi differential privacy [Mir17].

[7]Here we assume $\varepsilon > \rho$, which is the setting of interest.

[8]The entire discussion in this section applies equally well to the continuous analogues of these distributions.

[9]Note that this later condition is added to the algorithm to avoid double-counting the probability that $Z = 0$.

[10]We do compute $t = \lfloor \sqrt{\sigma^2} \rfloor + 1 = \inf\{n \in \mathbb{N} : n^2 > \sigma^2\}$ and count this as a single operation; this can be done exactly with rational operations (and binary search).