[Reviews · NeurIPS 2020]

Review 1

Summary and Contributions: This paper provides a variety of results relating to a discrete version of the Gaussian distribution that can be used for differential privacy applications with count data. The paper contains many slight improvements over other results in the literature and some unnecessary material (the comparison and discussion with the discrete Laplace does not add anything new or unexpected). Supplementary material, such as the Bernoulli sampling approach, can replace it to make the paper stronger and more self-contained. The sampling algorithm, even though the authors did not consider to be the strength of the paper, is the reason I think it should be accepted. A camera ready version of this paper should perform the replacement of content mentioned above and should include a better complexity analysis of the sampling algorithms. It should include complexity in terms of the values of the privacy parameters and/or the number of bits used to represent them (the result should be more informative than "constant").

Strengths: A variety of sampling algorithms for the binomial, discrete Laplace, and discrete Gaussian.

Weaknesses: Experimental comparisons with the discrete Laplace are not necessary, do not introduce any ideas that are not present in prior work, and use up valuable space that should be reserved for sampling algorithms. The expected running time needs to show complexity in terms of the privacy parameters or bits available to represent them.

Correctness: I believe so and checked most of the claims. I did not verify that the exact constants were correct.

Clarity: Yes

Relation to Prior Work: Yes

Reproducibility: Yes

Additional Feedback: These are comments following the author response. After reading the author response and thinking about the main points of the paper, I am even more supportive. I think this paper will have an impact. I view the paper as a user's guide to implementing a provably correct differential privacy system. Among the highlights are: - A provably correct version of the binomial with p defined as an exponential. - A provably correct version of the geometric mechanism. For both of these, the authors may not see much novelty but there is significant value in having these ideas together in one place. - A provably correct version of the discrete Gaussian. Apparently I value it more than the authors do. The following statement is significant to anyone who wants to implement it: "In contrast, Karney’s method still involves representing real numbers, but this can be carefully implemented on a finite computer using a flexible level of precision and lazy evaluation of the random bits." - A guide to how to use it with differential privacy. Proposition 23 is potentially useful for 1 discrete gaussian and an almost optimal conversion to approximate differential privacy. It could be useful in the main text. The authors should pay attention to strengthen the link to NeurIPS. A typical NeurIPS reader may not see the connections. These or other ideas are worth emphasizing in the paper: - If marginals and other statistics are published with discrete gaussian noise, they are much easier to analyze than with the discrete laplace. Practically speaking, there is no loss in the analyst pretending that it was pure gaussian noise - Some hypothesis testing frameworks for differential privacy have simple well-behaved noise asymptotic distributions when gaussian noise is used. There is no provably correct Gaussian implementation for DP but a provably correct discrete gaussian is good enough - Provably correct differentially private variations of SGD can make use of the discrete gaussian. For example, there is potential to make a variation of this paper: "Dissecting Adam: The Sign, Magnitude and Variance of Stochastic Gradients" (ICML 2018) which focuses on basing updates on the sign rather than magnitude.


Review 2

Summary and Contributions: This paper considers the basic problem of implementing the Gaussian mechanism while preserving differential privacy on finite precision device. The authors propose a discrete variant of the Gaussian distribution that while formally preserving differential privacy guarantees, also enjoys (asymptotically) the same utility guarantee as its continuous variant.

Strengths: 1. The problem is very important, and most deployed systems that use Gaussian mechanism, e.g, DP-SGD etc. ignores this issue completely. 2. The formulation of the discrete variant is natural, however proving privacy guarantees go through non-trivial fourier analytic techniques. 3. The paper gives an almost concrete characterization in terms of concentrated differential privacy, and approximate differential privacy.

Weaknesses: 1. While I liked the paper in terms of the spectrum of the results it provides, I am unsure about the main technical contribution of the paper. Most of the technical results seem to be a careful usage of techniques from prior work, e.g., Reg09. (The paper does faithfully acknowledge these clearly though.) I would like a more detailed description of the main technical ideas in the paper, which distinguishes it from a paper that has lengthy but monotonous calculations.

Correctness: I did not verify the correctness in details. Intuitively, it made sense to me.

Clarity: Most of the paper is well written, except the comment in the weakness. I am willing to increase my score if this issue is addressed in the rebuttal.

Relation to Prior Work: Yes.

Reproducibility: Yes

Additional Feedback:


Review 3

Summary and Contributions: This paper proposes a discrete Gaussian distribution and proves that the privacy and utility guarantees for adding noise drawn from it are the same as adding continuous Gaussian noise. It also provides a method for sampling from the discrete Gaussian distribution. Additionally, the paper provides a conversion from RDP to approximate DP which has a clean and numerically stable formula compared to previous optimal conversion.

Strengths: 1. The paper proposes and analyzes the discrete Gaussian distribution, which is important to DP since prior work has shown finite-precision approximation of continuous noise can cause privacy violation. 2. The paper provides a conversion from RDP to approximate DP with a clean and stable formula, which can be pretty useful in practice.

Weaknesses: I don't see any clear weakness of the paper.

Correctness: It seems correct to me.

Clarity: The paper is well written. I have two minor comments regarding the presentation. 1. Why is CDP defined specifically for 1/2*epsilon^2 but not generally for any \rho? I think some parts of the paper used \rho-CDP for \rho not equal to 1/2*epsilon^2, such as in Proposition 7. 2. Is it possible to provide some numerical comparison for the RDP to DP conversion with the previous results?

Relation to Prior Work: Yes.

Reproducibility: Yes

Additional Feedback:


Review 4

Summary and Contributions: Adding Gaussian or Laplacian noise to the value of a function evaluated on a sensitive  dataset is a commonly used way to guarantee privacy. However continuous noise can not be exactly represent by finite computers and some works (Mir12) has proved that naive numerical finite-precision approximations may damage privacy. Besides that adding continuous noise to discrete data makes the result less interpretable. Therefore this paper proposes the discrete Gaussian noise to the context of differential privacy(DP). Overall the authors demonstrate that the discrete Gaussian achieves the same level of privacy and utility as the continuous Gaussian. Besides that they provides an efficient sampling method of the discrete Gaussian. The sampling method is based on prior method (Kar16). The main contribution of the paper is the non-trivial proof of the privacy and utility of the discrete Gaussian mechanism.

Strengths: 1. Privacy, utility and efficiency are three important aspects of a differential privacy algorithm. This paper proves the three aspects of the discrete Gaussian mechanism. These proofs are not trivial. The Theorem 4 is the main result of the paper, which shows that the standard deviation of the added noise is proportional to the scale of the sensitivity of a query. 
 2. Figure 2 shows that the tail bounds and variance for continuous, discrete, and rounded Gaussians. The utility of the discrete Gaussian is better than rounding the continuous Gaussian, which may make the discrete Gaussian mechanism more favored. 
 3. The paper proves the tight approximate differential privacy bound for discrete Gaussian. 
 4. The authors give a detailed Broader Impact, which is informative.

Weaknesses: 1. In my opinion discrete Gaussian mechanism proposed by this paper is not a big innovation, since the discrete Laplace distribution has  been introduced to  DP in (GRS12) and be used in the 2020 US Census. Applying discrete Gaussian noise to DP is a natural derivational work of the discrete Laplacian noise to DP. 2. In Section3.1 the authors give a thorough comparison between the discrete Gaussian and discrete Laplace. The conclusions deduced by the discrete distribution are almost the same as the prior conclusions of the continuous distribution. 3. This paper does not compare their methods with the exponential mechanism(Frank McSherry and Kunal Talwa 2007). Exponential mechanism can also output a discrete answer for a query.

Correctness: Yes, the claims and method seem correct. This paper lacks empirical methodology. It is a theoretical paper.

Clarity: Yes, the paper is clearly written.

Relation to Prior Work: Yes, it is discussed.

Reproducibility: Yes

Additional Feedback:


Review 5

Summary and Contributions: This paper proposes a discrete Gaussian mechanism for differential privacy, motivated by the fact that the (continuous) Gaussian mechanism may lead to privacy errors when implemented in floating point. The mechanism proposed is what one should expect -- basically (but not exactly) rounding a continuous Gaussian distribution. The Gaussian mechanism possesses some good privacy properties -- such as ease of composition; analogous properties of the discrete mechanism are established.

Strengths: + The paper is highly technical and in particular the properties are rather challenging to prove.

Weaknesses: - The paper could have done a better job of motivating the problem. It is true that the lower order bits in a floating point representation might leak information, but it's unclear how many of these bits are actually revealed to any practical adversary. Even so: why doesn't a simple randomized rounding at the end of a group of computations work? Is it because the difference accumulates? For an iterative method, how does the difference between a perfect and an imperfect implementation accumulate with rounds? The paper would be a lot stronger if these points were clarified. - The proofs while highly technical and nice do not involve a lot of novel ideas. But overall this is a minor weakness I think.

Correctness: The claims appear correct.

Clarity: Mostly well-written but a little dense.

Relation to Prior Work: The paper could do with some more discussion of the prior work and why it fails. Another work that is relevant but not quite the same is the staircase mechanism (https://arxiv.org/abs/1212.1186) which also has some optimality properties.

Reproducibility: Yes

Additional Feedback:

[Author Response · NeurIPS 2020]

We thank all three reviewers for their comments and suggestions.

**Reviewer 1:**

*Discrete Laplace comparison:* We consider the comparison with prior related work to be important, as it
highlights the value of our discrete Gaussian over the discrete Laplace. This comparison is crucial to help the
reader/practicioner(/reviewer) weigh the pros and cons of each method. Unfortunately, the advantages of the
Gaussian over the Laplace are not as widely known as they should be, so we consider it worthwhile to reiterate
this. Nonetheless, we will consider reallocating space in the final version.

*Runtime analysis:* The running time of our algorithm can be shown to be subexponential (measured by
the number of arithmetic operations, where the bit complexity of each operation is determined by the bit
complexity of the input parameters). We will add a statement on this to the final version of the main document.
(Currently there is a brief discussion in the supplement, which we will flesh out into a full proof.)

*Sampling Algorithm:* We like our algorithm and consider it to be a contribution, as it is simpler than the prior
work by Karney. However, we chose not to emphasize it because there is prior work; instead we emphasized
our other contributions which are more clearly novel. We also did not want to dedicate space to a comparison
with Karney's algorithm.

**Reviewer 2:**

*Technical content:* The technical contribution of the paper is to solve the problem of practical implementation
of differentially private noise addition. We combine a number of techniques to solve the problem. We agree
that some of the techniques that we apply, such as the Poisson summation formula, are known within certain
communities. However, to the best of our knowledge, these techniques are not well known to the NeurIPS and
Privacy communities, and have not been applied before in this context. Furthermore, we believe that all of
our lemmata and theorems are novel and fundamental statements. Additionally, the focus of the cryptography
community is specifically on the high-dimensional discrete Gaussian, which is crucially believed to be *hard*
to sample as the dimension grows, a desirable feature for cryptographic applications; while we rely on the
efficient sampling for the univariate case. In that sense, the viewpoint and goals are fundamentally different.

**Reviewer 3:**

*Parameterization of CDP:* The parameterizations $\rho$-CDP and $\frac{\varepsilon^2}{2}$-CDP are interchangeable. We prefer the latter
as it puts CDP on the same familiar "scale" as pure $\varepsilon$-DP and approximate $(\varepsilon, \delta)$-DP (indeed, $\varepsilon$-DP implies
$\frac{\varepsilon^2}{2}$-CDP which in turn implies $(O(\varepsilon\sqrt{\log(1/\delta)}), \delta)$-DP) – i.e., people are more accustomed to $\varepsilon$ as a privacy
parameter than $\rho$. We revert to $\rho$ when we are comparing CDP with approx DP as otherwise the parameter $\varepsilon$
would be overloaded. We will make this correspondence of $\rho = \frac{\varepsilon^2}{2}$ more explicit in the final version.

*Numerical comparison of RDP to DP conversion with previous results:* We did not include the comparison
because the numerical instability issues created weird artifacts in our plots. The code for the plots is included in
the supplementary material and this comparison can be added to the plot by simply changing `numeric=False`
to `numeric=True` at the end of the file and re-running the code. The resulting plots are included below.

Figure 1: Numerical comparison of RDP to DP conversion with previous results

[Meta-Review · NeurIPS 2020]

The paper introduces the Discrete Gaussian mechanism for differential privacy. The results can be applied to make sure that the no information is leaked in implementations of differentially private algorithms. The authors are urged to make a better connection to the NeurIPS community.